



# On the uncertainty of initial condition and initialization approaches in

# variably saturated flow modeling

Danyang Yu[1], Jinzhong Yang[1*], Liangsheng Shi[1], Qiuru Zhang[1], Kai Huang[2], Yuanhao Fang[3], Yuanyuan Zha[1*]

[1]State Key Laboratory of Water Resources and Hydropower Engineering Sciences, Wuhan University, Wuhan, Hubei 430072, China

[2] Guangxi Hydraulic Research Institute, Nanning, Guangxi 530023, China

[3] School of Earth Sciences and Engineering, Hohai University, Nanjing 210098, China

* *Corresponding author*: Jinzhong Yang (jzyang@whu.edu.cn); Yuanyuan Zha (zhayuan87@whu.edu.cn)





**Abstract:**

Soil water movement has direct effects on environment, agriculture and hydrology. Simulation of soil water movement requires accurate determination of model parameters as well as initial and boundary conditions. However, it is difficult to obtain the accurate initial soil moisture/matric potential profile at the beginning of simulation time, making it necessary to run the simulation model from arbitrary initial condition until the uncertainty of initial condition (UIC) diminishes. The behavior of this process is usually defined as "warming up". In this paper, we compare two commonly used methods for quantifying the UIC (one is based on running a single simulation recursively across multiple hydrological years, and the other is based on Monte-Carlo simulations with various initial condition) and identify the "warm-up" time $t_{wu}$ (minimum time required for the model to warm up to eliminate the UIC) required with different soil textures, meteorological conditions, and soil profile lengths. Then we analyze the effects of different initial conditions on parameter estimation within two data assimilation frameworks (i.e, ensemble Kalman filter and iterative ensemble smoother) and assess several existing model initializing that uses available data to retrieve initial soil moisture profile. Our results reveal that Monte-Carlo simulations and the recursive simulation over many years can both demonstrate the temporal behavior of UIC and a common threshold is recommended to determine the warm-up time for both methods. Moreover, the relationship between warm-up time for variably saturated flow modeling and the model settings (soil textures, meteorological conditions and soil profile length) are quantitatively identified. In addition, we propose a "warm-up" period before assimilating data in order to obtain a better performance for parameter and state estimation.

**Key words:** Variably saturated flow; Initialization methods; Initial condition uncertainty; Data assimilation; Soil moisture



## 1. Introduction

Understanding the movement of soil water is of great importance due to its direct effects across different disciplines, such as in environment, agriculture and hydrology (Doussan et al., 2002). However, the modeling of flow in variably saturated soil is complicated by many difficulties, including highly
variable and nonlinear physical processes, as well as limited information about the soil hydraulic properties, initial conditions, and boundary conditions (DeChant, 2014; Rodell et al., 2005; Seck et al., 2014; Bauser et al., 2016; Li et al., 2012). The soil hydraulic parameter uncertainty is identified as one of the major uncertainty sources in vadose zone hydrology and many studies have been focused on this topic. A highly relevant research area, inverse problem, has been developed to reduce the uncertainty of
parameter by incorporating observational data (Erdal et al., 2014; Montzka et al., 2011; Wu and Margulis, 2013; Wu and Margulis, 2011). Initial and boundary conditions also introduce uncertainty during the simulation of soil water flow (Ataie-Ashtiani et al., 1999; Forsyth et al., 1995; Szomolay, 2008). For instance, the uncertainty of boundary conditions, due to flawed/noise-contaminated meteorological data or fluctuating groundwater table, has been investigated in the past (Freeze, 1969; French et al., 1999; van
Genuchten and Parker, 1984; Ji and Unger, 2001; Xie et al., 2011).

Many publications have addressed the issue of the uncertainty of initial condition (UIC) in modeling soil water movement. For example, Walker and Houser (2001) compared the simulation with degraded soil moisture initial condition to that with true initial condition and found the discrepancy did not fade away even after one month. Then, Mumen (2006) concluded that the initial soil water state was one of
the most important factors for estimating soil moisture in the case of bare soil. Chanzy et al. (2008) tested three initial water potential profiles and found that initialization had a strong impact on the soil moisture prediction. All these studies showed that the incorrect initial condition may lead to false results. Based on the availability of information, initial condition can be obtained from different initialization approaches , e.g., an arbitrary uniform profile (Chanzy et al., 2008; Das and Mohanty, 2006; Varado et al, 2006), a
linear interpolation with situ observation (Bauser et al., 2016), a steady-state soil moisture profile induced with a constant infiltration flux (Freeze, 1969). All of the approaches involve great uncertainties due to nonlinearity of soil moisture profile, observation error, or inaccurate boundary condition. As a result, it is crucial to explore the effects of UIC on model outputs and compare the uncertainties inherited from



various initialization approaches.

To minimize the bias introduced by initial states and attain more accurate model results in earlier run, a common approach of model initialization is to obtain initial condition inherited from the warm-up model with preceding meteorological data. Starting from an arbitrary initial condition, this approach runs the model using a certain period of meteorological data ($t_{wu}$) until the model state (e.g., soil moisture) reaches an equilibrium state, which is defined as the state when UIC is negligible during simulation. The

equilibrium state can be obtained by either running Monte-Carlo simulations until the states from different initial conditions converge to the same value (hereafter referred as Monte-Carlo method) (Chanzy et al., 2008), or running a single simulation for several years by repeating one-year or multiple-year meteorological condition until the state at an arbitrary date cease to vary from year to year (Spin-up method) (Dechant and Moradkhani, 2011; Seck et al., 2014). To the best of our knowledge, there is no

comparison made between these two methods to date. Moreover, the determination of warm-up time $t_{wu}$ is key to the success of this approach (Ajami et al., 2014; Rahman and Lu, 2015). An underestimation of $t_{wu}$ may bring uncertainty from arbitrarily-specified initial condition prior to initialization, while a large $t_{wu}$ for initialization leads to higher computational demands (Rodell et al., 2005). A variety of modeling settings, such as soil hydraulic properties, meteorological conditions, and soil profile lengths, have strong

influences on $t_{wu}$ (Ajami et al., 2014; Cosgrove et al., 2003; Lim et al., 2012a; Walker and Houser, 2001). Thus, the determination of $t_{wu}$ should be investigated thoroughly with different settings.

        Data assimilation has become a popular tool in the field of vadose zone hydrology (Vereecken et al., 2010; Chirico et al., 2014; Medina et al., 2014a, 2014b). Some studies showed that the poor initial soil moisture profile can be corrected by assimilating near-surface measurements (Galantowicz et al., 1999;

Walker and Houser, 2001; Das and Mohanty, 2006). Oliver and Chen (2009) discussed several possible approaches to improve the performance of data assimilation through improved sampling of the initial ensemble, and suggested the use of the pseudo-data. Recently, Tran et al. (2013) found that the decreasing assimilation interval could improve the soil moisture profile results induced by wrong initial condition and Bauser et al. (2016) has addressed the importance of UIC in data assimilation framework.

Nevertheless, investigation of the influence of UIC on data assimilation results (i.e., parameter and state estimation) is limited. For instance, during data assimilation, the initial ensemble are often assumed to be





known without uncertainty (Shi et al., 2015) or created by adding Gaussian noise to the initial estimate (Huang et al., 2008), both of which may result in false outputs. Currenlty, related researches are all based on a sequential data assimilation approach (i.e., ensemble Kalman filter, or EnKF (Walker and Houser,

2001; Oliver and Chen, 2009)), which incorporates observation in a sequential fashion, so the effect of UIC can be eliminated quickly. Compared to EnKF, an iterative ensemble smoother (IES), which assimilates all data available simultaneously, can obtain reasonably good history-matching results and performs better in strongly nonlinear problems (Chen and Oliver, 2013). However, IES utilize all the observation simultaneously at every iteration and UIC may have a more persistent effect on IES. Thus, it

is important to understand the propagation process of UIC during variably saturated flow modeling, to identify the warm-up time $t_{wu}$ under a variety of scenarios, and to compare different existing initialization methods within various data assimilation frameworks.

       The objectives of this paper, therefore, are to: a) compare the temporal evolution of UIC with two common methods (Spin-up method and Monte-Carlo method) and identify the warm-up time $t_{wu}$ under

different soil hydraulic parameters, meteorological conditions and soil profile lengths; b) analyze the effects of different initial conditions on parameter estimation during data assimilation with EnKF or IES, and c) propose a selection scheme for choosing a suitable approach of initializing variably saturated flow models within different data assimilation frameworks to minimize the influence of UIC. We first summarize the governing equations of variably saturated flow and method of UIC quantification in

Section 2. Then we present results of synthetic simulations designed to investigate the propagation of UIC under different scenarios in Section 3, which is complemented by results for field data in Section 4. Finally, we present our conclusions in Section 5.

## 2. Method

### 2.1 One-dimensional soil water movement

Richards' equation can be used to describe the one-dimensional, vertical soil water movement, which is given as:

$$\frac{\partial \theta}{\partial t} = \frac{\partial}{\partial z}\left[ K\left( \frac{\partial h}{\partial t} + 1 \right) \right] \tag{1}$$

where $h$ [$L$] represents the pressure head; $\theta$ [-] denotes volumetric soil moisture; $t$ [$T$] indicates the time;





$z$ [$L$] is the spatial coordinate taken positive upward; $K$ [$LT^{-1}$] denotes the unsaturated hydraulic

conductivity. The solution of one-dimensional Richards's equation is numerically solved by a noniterative

numerical scheme, which was originally proposed in (Ross, 2003; Ross, 2006). By using the primary

variable switching scheme, this scheme uses the soil moisture as the unknown variable for unsaturated

nodes and pressure head for saturated nodes (Zha et al., 2013). It can greatly reduce the computation cost

of variably saturated flow modeling in soils under atmospheric boundary condition, where alternative

drying-wetting conditions are often encountered.

To obtain the solution of Eq. (1), the knowledge of functions $K$ and $\theta$ versus $h$ must be required. In

this study, we use the van Genuchten-Mualem model (van Genuchten, 1980; Mualem, 1976) to describe

the relationships,

$$\theta(h) = \theta_r + \frac{\theta_s - \theta_r}{\left[1 + |\alpha h|^n\right]^m} \qquad (2)$$

$$K(\theta) = K_s S_e^{0.5}[1 - (1 - S_e^{1/m})^m]^2 \qquad (3)$$

where $K_s$ [$LT^{-1}$] denotes the saturated hydraulic conductivity; $\theta_s$ and $\theta_r$ represent the saturated and

residual soil moistures, respectively; parameters $\alpha$ [$L^{-1}$] and $n$ are related to the measure of the pore-

size density functions and $m=1-1/n$ ($n>1$); the effective saturation degree $S_e$ is defined as

$S_e = (\theta - \theta_r)/(\theta_s - \theta_r)$.

Initial and boundary conditions are needed to solve the one-dimensional Richards' equation. The

initial condition could be the states of soil moisture

$$\theta(z,t)\big|_{t=0} = \theta_0(z) \qquad (4)$$

where $\theta_0(z)$ is the initial soil moisture profile.

The upper boundary condition is specified as atmospheric boundary (state-dependent boundary

condition (Šimůnek et al., 2013)) in this study,

$$|q| = \left| -K\frac{\partial h}{\partial z} - K \right| \le \left| E_p - P_p \right| \qquad (5)$$

$$h_m > h > h_c \qquad (6)$$




where $q$ [$LT^{-1}$] is the Darcian flux at the soil surface; $E_p$ [$LT^{-1}$] denotes the potential evaporation; $P_p$ [$LT^{-1}$] represents the precipitation intensity; $h_m$ [$L$] and $h_c$ [$L$] are maximum and minimum pressure heads

allowed at the soil surface, respectively. The value of $h_m$ is set to 0, whereas $h_s$ is detemined as -100 m.

The bottom boundary condition is the free drainage boundary:

$$\left.\frac{\partial h}{\partial z}\right|_{z=z_N} = 0 \tag{7}$$

where $z_N$ is the depth of bottom boundary.

**2.2 UIC quantification**

The investigation of uncertainty in this study includes model states (e.g., soil moisture) and model parameters, where the uncertainty of initial condition (state at $t$=0) is a special case of state uncertainty. We consider two cases in our analysis. First, we consider a particular situation when UIC is the only uncertain source and all the model parameters are known. In this case, the temporal decay of UIC (either with or without observation) can be clearly demonstrated. Second, a more complex situation, including

both uncertain initial condition and model parameters, is considered during the data assimilation of soil moisture observation. Data assimilation methods, i.e., EnKF and IES, are used to update the model parameters and state simultaneously when observation data are available.

2.2.1 The indexes of Spin-up and Monte-Carlo methods

The uncertainty of initial condition can be measured by the percent change $PC$ for Spin-up method

(Ajami et al., 2014; Seck et al., 2014) or the ensemble spread $S_p$ for Monte-Carlo method (Reichle and Koster, 2003). Percent change is calculated as:

$$PC = 100\left|\frac{M_1 - M_2}{M_2}\right| \tag{8}$$

where $M_1$ is the monthly mean of soil moisture from the previous year and $M_2$ is the monthly average for the current year (de Goncalves et al., 2006).

The ensemble spread ($S_p$) is an useful index to quantify the difference of realizations in Monte Carlo simulation, and it is given as:

$$S_p(k) = \sqrt{\frac{1}{N}\sum_{i=1}^{N} Var(y_{i,k})} \tag{9}$$



with

$$Var(y_{i,k}) = \frac{1}{N_e - 1} \sum_{j=1}^{N_e} (\mathbf{y}_{i,j,k}^a - \langle \mathbf{y}_{i,k}^a \rangle)^2 \tag{10}$$


where $Var(y_{i,k})$ denotes the nodal ensemble variance at time $t_k$; $\mathbf{y}_{i,j,k}^a$ is nodal soil moisture value;

$\langle \mathbf{y}_{i,k}^a \rangle$ is the ensemble mean of $\mathbf{y}_{i,j,k}^a$; $i = 1, 2, …, N$ are the nodes of interest (can be part of the profile);

$j=1, 2, …, N_e$ is the ensemble number index; $N_e$ is the ensemble size, which is taken as 300 in this study

based on sensitivity analysis of the ensemble size on the calculated results. $S_p$ is the ensemble spread,

which can be thought as a square root of averaged variance over all interested nodes. When $N = 1$, the

concept of $S_p(k)$ is equivalent to the standard deviation of $\mathbf{y}_k^a$ at one location and time $t_k$.

2.2.2 Data assimilation approaches

   We employ EnKF and IES as the data assimilation approaches in this study. Fig. 1 illustrates the basic

ideas and differences of the two methods.

EnKF approach was first proposed by Evensen (1994) and has been widely used in variably saturated

flow problems (Huang et al., 2008; De Lannoy et al., 2007). This approach is a sequential data

assimilation (as shown in Fig. 1(a)) which incorporate observations into the model in order.

   In this work, we assume that hydraulic parameters $K_s$, $\alpha$, and $n$ are unknown, while the other

parameters $\theta_r$ and $\theta_s$ are supposed to be deterministic. The vector of parameter and state can be described

as,

$$\mathbf{y}_k = \left[ \mathbf{m}_k, \mathbf{u}_k \right]^T \tag{11}$$

where $\mathbf{m}_k$ is the parameter vector (i.e., $K_s$, $\alpha$, and $n$), $\mathbf{u}_k$ are state variables (i.e., pressure head and

soil moisture) at time $t_k$, the dimension of $y_k$ is $N_y$: $N_y = N_m + 2 N_d$, where $N_m$ indicates the amount of the

parameters to be estimated; $N_d$ are the number of nodes of the numerical model. The observation vector

can be defined as,

$$\mathbf{d}_{j,k} = \mathbf{d}_k + \mathbf{\varepsilon}_{jk} \tag{12}$$

where $\mathbf{d}_k$ denotes the observation at time $t_k$; $\mathbf{\varepsilon}_{jk}$ $(j=1, 2, …, N_e)$ are independent Gaussian noises added





to the observations; $\mathbf{d}_{j,k}$ is the observation vector for ensemble index $j$ at time $t_k$. Based on the differences of model forecast and observations, the state-parameter vector can be updated as:

$$\mathbf{y}_{j,k}^a = \mathbf{y}_{j,k}^f + \mathbf{K}_k(\mathbf{d}_{j,k} - \mathbf{H}\mathbf{y}_{j,k}^f) \tag{13}$$

where $\mathbf{y}_{j,k}^f$ denotes the estimated or initially guessed values of parameter and state, while $\mathbf{y}_{j,k}^a$ is the updated estimates; $\mathbf{H}$ is an observation operator, linking the relationship between the state-parameter vector and the observation vector. $\mathbf{K}$ represents the Kalman gain matrix, which can be calculated as,

$$\mathbf{K}_K = \mathbf{C}_k^f \mathbf{H}^T [\mathbf{H}\mathbf{C}_k^f \mathbf{H}^T + \mathbf{C}_{D_K}]^{-1} \tag{14}$$

where $\mathbf{C}_{D_k}$ indicates the covariance matrix of observed data errors, while $\mathbf{C}_k^f$ is the error covariance matrix of forecast ensemble, given by

$$\mathbf{C}_k^f \approx \frac{1}{N_e - 1} \sum_{j=1}^{N_e} \left\{ \left[ \mathbf{y}_{j,k}^f - \left\langle \mathbf{y}_k^f \right\rangle \right] \left[ \mathbf{y}_{j,k}^f - \left\langle \mathbf{y}_k^f \right\rangle \right]^T \right\} \tag{15}$$

where $\left\langle \mathbf{y}_k^f \right\rangle$ is the ensemble mean of $\mathbf{y}_k^f$.

Compared to EnKF, IES gives a better estimate by taking all the available observation into
consideration (van Leeuwen and Evensen, 1996), as presented in Fig. 1(b). Thus, it can keep the overall consistency of parameters and state variables over time effectively and has been increasingly used to solve the parameter estimation problem in hydrology (Crestani et al., 2013; Emerick and Reynolds, 2013). By calculating iteratively, the nonlinear relationship between observation and parameter is linearized and the information content of the observations can be fully utilized (Chen and Oliver, 2013). In this case, we
write the analyzed vector of model parameters $\mathbf{m}_j^r$, as

$$\mathbf{m}_j^{r+1} = \mathbf{m}_j^r + \mathbf{K}^r(\mathbf{d}_j^r - \mathbf{H}\mathbf{m}_j^r) \tag{16}$$

The notation is similar to the one presented for EnKF, where $r$ is the iteration index; $\mathbf{m}_j^r$ is the initially guessed or estimated parameters for realization $j$ at iteration $r$; $\mathbf{m}_j^{r+1}$ is the updated estimates for realization $j$ by conditioning on the observed information at iteration $r$. It should be noted that the $\mathbf{d}_j^r$
and $\mathbf{H}\mathbf{m}_j^r$ denotes the total number of observation and predicted data at iteration $r$, which is different





from EnKF. The Kalman gain $\mathbf{K}$ is defined as,

$$\mathbf{K}^r = \mathbf{C}_r^f \mathbf{H}^T [\mathbf{H}\mathbf{C}_r^f \mathbf{H}^T + \mathbf{C}_D + \lambda \text{diag}(\mathbf{H}\mathbf{C}_r^f \mathbf{H}^T)]^{-1} \qquad (17)$$

where $\mathbf{C}_r^f \mathbf{H}^T$ is the cross-covariance matrix between the prior vector of model and the vector of predicted data at iteration $r$; $\mathbf{H}\mathbf{C}_r^f \mathbf{H}^T$ is the auto-covariance matrix of predicted data at iteration $r$ and $\mathbf{C}_D$ is the

covariance matrix of observed data errors. $\lambda$ donates a dynamic stability multiplier, and diag $(\mathbf{H}\mathbf{C}_r^f \mathbf{H}^T)$ is a diagonal matrix with the same diagonal elements as $\mathbf{H}\mathbf{C}_r^f \mathbf{H}^T$. Mathematically, the dynamic stabilizer term facilitates the solution switching between the Gauss-Newton solution and the steepest-descent method, which is known as the Levenberg-Marquardt approach (Pujol, 2007).

## 3. Numerical examples

A series of synthetic numerical experiments are performed in this section. In Section 3.1, we give a general description of the numerical experiments. In order to gain a better understanding of the propagation of the UIC, all the hydraulic parameters (i.e., $K_s$, $\alpha$ and $n$) are assumed to be deterministic and the UIC is the only uncertainty source in Section 3.2. Finally, the numerical cases are designed to evaluate performances of data assimilation algorithms initialized by various initial conditions in Section

3.3, in which the parameter uncertainty is taken into consideration in conjunction with UIC.

### 3.1 General description of model inputs

As shown in Table 1, there are five common methods to perscribe initial conditions in variably saturated flow model based on the availablitity of information, including a uniform 50% relative saturation over the entire soil profile (hereafter abbreviated as IC-HfSatu) (Margulis et al., 2002), a linear

interpolation between observations at the beginning of simulation (IC-ObsInt) and a steady-state soil moisture profile with a constant infiltration flux (IC-Flux). In this study, the flux is set as 1 mm/d. Besides, we also employ two warm-up methods, which give initial conditions by running the model prior to the beginning of simulation period with available meteorological data (as shown in Fig. 1). If the meteorological data before the simulation period is available, it is used in the warm-up method to obtain

the initial condition (IC-WUP); otherwise, we use the meteorological data at the simulation period (IC-WUE) as a surrogate. Starting with guessed parameter and initial condition (i.e., we generate two

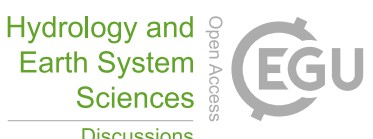

ensembles with guessed means and variances, whereas the means may be biased from the true values),
IC-WUP or IC-WUE first warms up the model with available meteorological data, and then uses the
ensemble of soil moisture profiles (still uncertain due to the uncertainty of parameters prior to data
assimilation, while the bias and uncertainty introduced by state at $t_{pre}$ are reduced) on the last warm-up
day (or the beginning of the simulation time $t_0$) as the initial condition of the formal simulation/data
assimilation (see Fig. 1). The length of warm-up time $t_{wu}$ is determined based on the conclusion in Section
3.2.2(a).

Four typical types of soil (Sand, Loam, Silt and Clay loam) are chosen in this study to explore the
impacts of soil hydraulic property on UIC. Table 2 lists the mean values of uncertain hydraulic parameters
(i.e., $K_s$, $\alpha$ and $n$) according to Carsel and Parrish (1988).

To investigate the impact of model settings on the temporal evolution of UIC, three different
meteorological conditions are employed. M-AC, M-SC and M-HC in Fig. 1 represent three precipitation
and potential evaporation data from three different regions (arid region, semi-arid region and humid
region) in China.

Unless otherwise specified, the soil profile is set to be 300 cm thick and is filled with Loam. The flow
domain is discretized into 60 grids with a grid size of 5 cm. Besides, the total simulation time during the
synthetic simulation is one year (365 days). In addition, the default value of the upper boundary condition
is set to be M-SC. Other specifications and assumptions for our model simulation runs are given in Table
250  1.

### 3.2 The temporal evolution of UIC

3.2.1 Comparison of UIC quantification methods

A synthetic experiment is conducted to compare two methods (i.e., Spin-up method and Monte-Carlo
method) in quantifying UIC. Using the spin-up method, the first case runs a single simulation for 10 years
by repeating the preceding meteorological condition starting with IC-HfSatu (Fig. 3(a)), and the
percentage cutoff PC is calculated. In the second case, the Gaussian noise with a standard deviation of 3%
is added to the IC-HfSatu to generate an ensemble with different initial soil moisture profiles. Then we
run the model until the ensemble spread $S_p$ is small (Fig. 3(b)). Finally, the PC and $S_p$ values of the two
cases versus time are compared in Fig. 3(c).





260        As shown in Fig. 3(a), there is a visible difference bewteen the monthly-averaged soil moistures at the beginning and the 12$^{th}$ months, while the difference is much smaller for $\theta$ at the 12$^{th}$ and 24$^{th}$ months, indicating the decay of UIC. Similarly, the soil moistures for different realizations gradually get closer to each other, implying the decay of UIC over simulation time. As shown in Fig. 3(c), PC and $S_p$ values gradually decrease with the simulation time, and their values are approximately the same after $t>6$ months.

The significant difference at the beginning (PC of 4.7% and $S_p$ of 2.6%) is due to different initial soil moistures given by the Spin-up and Monte-Carlo methods. The result indicates that the widely-used Spin-up method and Monte-Carlo method are equivalent to one another in terms of quantifying UIC. We will use Monte-Carlo method for the rest of the study since it is consistent with the data assimilation approaches used in this study.

270        The determination of the threshold value when UIC is regarded to have negligible effect on prediction has been discussed in previous studies (Ajami et al., 2014; Lim et al., 2012; Seck et al., 2014). In general, $PC$ or $S_p$ values of 1% (Yang et al., 1995), 0.1% (de Goncalves et al., 2006), or 0.01 % (Henderson-Sellers et al., 1993) have been used as threshold for the model. In this study, we recommend a threshold of 0.5% for both Spin-up and Monte-Carlo methods, and the corresponding warm-up time $t_{wu}$ is 8 months, which

is sufficiently long for UIC to diminish and the difference between $PC$ and $S_p$ to become small.

3.2.2 The influencing factors on UIC

       The Monte-Carlo method is used in this part to obtain the warm-up time $t_{wu}$ and a number of scenarios are constructed under a variety of conditions (different soils, different meteorological conditions and soil profile lengths). First, the influence of soil texture and meteorological condition on $t_{wu}$ are examined. Four

different types of soils (Sand, Loam, Silt and Clay loam listed in Table 2) under three typical meteorological conditions (M-AC, M-SC and M-HC) are employed in these scenarios, while the other model inputs use the default values (see Table 1). Besides, the influence of different soil profile lengths (1 m, 3 m, 5 m, 10 m, 15 m, and 20 m) on UIC is also investigated.

*a.* The influences of soil texture and meteorological condition

285        Fig. 4 plots $t_{wu}$ with four different soils under three typical meteorological conditions. The computational times vary greatly according to soil property. We find that $t_{wu}$ of Sand are all less than one day, whereas $t_{wu}$ of Loam are 412 days, 242 days, and 195 days respectively. In addition, the warm-up



times of Silt and Clay loam with M-AC and M-SC exceed 10 years, while those with M-HC are 264 days

and 253 days. The results imply that the warm-up time $t_{wu}$ for the fine-textured soil is larger than that for

coarse-textured soil. To further explore the reason behind, results of soil moisture profiles at the given

days (e.g., days 1st and 30th) for four kinds of soils are manually inspected (results not shown). For Sand,

due to its fast drainage property, the soil moisture ensemble converges extremely quickly and most of the

values at the profile are maintained as residual soil moisture. Thus, the UIC of sand disappears very fast.

On the other hand, the soil moisture results for Silt and Clay loam show slow convergence. For Loam,

the convergence speed is smaller than that of Sand but larger than that of Clay loam. Therefore, faster

drainage property leads to smaller warm-up time.

In addition, the meteorological condition has a strong impact on UIC and $t_{wu}$ decreases with the

increase of precipitation. For example, with soil Loam, the order of $t_{wu}$ is M-HC<M-SC<M-AC.

Regarding Silt and Clay loam, $t_{wu}$ of M-AC and M-SC decrease from more than 10 years to 264 days and

253 days with a humid climate M-HC, respectively. With intensive and excessive rainfall events, $\theta$

approaches to the saturated soil moisture, leading to a sudden drop of $S_p$. Thus, the meteorological

condition, especially the precipitation, plays an important role in the propagation of UIC. Excessive and

intensive rainfall could greatly reduce the impact on simulation solely induced by UIC (i.e., no parameter

uncertainty).

*b. The influence of soil profile length*

To investigate the effects of soil profile length on warm-up time, we investigate the the $t_{wu}$ values for

simulations with various soil profile lengths. As presented in Fig. 5, the $t_{wu}$ for soil lengths of 1 m, 3 m, 5

m, 10 m, 15 m and 20 m are 0.11 year, 0.57 year, 0.74 year, 1.57 years, 2.78 years and 4.3 years

respectively, indicating that the warm-up time increases with increasing depth of soil column. This can

be attributed to a larger portion of soil is not affected by boundary conditions and UIC decays more slowly.

We also examine the case for substituting free drainage boundary for a prescribed groundwater table. The

results indicate that the $t_{wu}$ is further shortened due to the restriction of bottom saturation condition (not

shown).

In addition, $t_{wu}$ in homogeneous loam reveals a power law relationship with the length of soil profile.

According to the fitted curve in Fig. 5, the warm-up time $t_{wu}$ is more than seven years for a depth $d$ of 30





m (e.g., North China Plain, (Huo et al., 2014)) and 700 years for $d$=1000 m (e.g., Yucca Mountain Site, (Flint et al., 2001)) with loam soil. This result suggests that we should be very careful to deal with simulation with a long unsaturated profile, where the UIC can exist an extremely long time and influence the simulation/data assimilation results.

**3.3. Initialization of data assimilation**

In this sub-section, five initialization methods (IC-HfSatu, IC-ObsInt, IC-NetFlux, IC-WUP and IC-WUE) are assessed to investigate the effect of initial condition on model state and parameter estimations within two data assimilation frameworks (EnKF and IES). The initial realizations of soil hydraulic parameters $K_s$, $\alpha$ and $n$ are generated following logarithm normal distributions, with mean values of 4.7

md$^{-1}$, 8.6 m$^{-1}$ and 1.8, and variances (log-transformed) of 0.1, 0.3 and 0.006. Compared with the reference values ($K_s$, $\alpha$ and $n$ for Loam are 0.2496 md$^{-1}$, 3.6 m$^{-1}$ and 1.56) listed in Table 2, the prior means of unknown parameters are biased.

3.3.1 General description of the cases for various initialization methods

Several test cases are conducted to explore the effects of initialization on parameter estimation under

various data assimilation frameworks. Cases 1 and 2 investigate the effects of five initialization methods (Table 1) on individual parameter estimation with EnKF and IES, respectively. Furthermore, to find out the role of initial condition in multi-parameter inverse problems, Case 3 and Case 4 are conducted to estimate $K_s$, $\alpha$ and $n$ simultaneously. To explore the impact of assimilation time on parameter estimation with IES, Case 5 is implemented with a simulation time of 60 days. It should be noted that the warm-up

methods (IC-WUP and IC-WUE) used in IES warms up model before every iteration (as presented in Fig. 1(b)), since the initialization of IES by warming up the model for only the first iteration leads to poor assimilation results.

The synthetic observations used for data assimilation are generated by running the model with "true" parameter (Loam) and "true" initial condition (produced by warming up model with a sufficient long $t_{wu}$

=10 years). The generated observations are perturbed by observation errors (a standard deviation of 1%), which are assumed to be Gaussian. In addition, the observation at 10 cm is assimilated into model every 10 days. The details of the model inputs for Case 1 to Case 5 are listed in Table 3.





### 3.3.2 Quantitative index

To quantify the quality of model parameter and state estimations, root mean square of estimated

parameters ($RMSE_m$) and soil moisture ($RMSE_{obs}$) are computed as follows:

$$RMSE_m = \sqrt{\frac{1}{N_e}\sum_{j=1}^{N_e}(P_j^E - P^T)^2} \tag{18}$$

$$RMSE_{obs} = \sqrt{\frac{1}{N_{obs}}\sum_{n=1}^{N_{obs}}(x_n - x_n^{obs})^2} \tag{19}$$

where $P_j^E$ represents the parameter estimation of realization $j$ on the last simulation day; $P^T$ represents

the true parameter listed in Table 2. $N_e$ is the total number of realizations. $x_n$ and $x_n^{obs}$ indicate the

predicted and measured soil moistures, respectively. $N_{obs}$ is the amount of observations.

### 3.3.3 Result comparison

The results for parameter estimation ($\ln K_s$) using the two data assimilation frameworks and under the

various initialization methods are compared in Fig. 6. In Fig. 6(a), the estimated $\ln K_s$ values of EnKF are

presented. In general, the $\ln K_s$ estimations under different initial conditions all gradually approach the

"true" values with assimilation time, but the final assimilation results are different. For IC-HfSatu,

because the initial profile is uniform and arbitrarily specified, the assimilation results are both affected by

the parameter uncertainty and UIC. Thus, the decreasing of $RMSE_m$ is the slowest and the final parameter

estimation result is the worst. In contrast, the initial conditions generated by warm-up methods (IC-WUP

and IC-WUE) can eliminate the UIC in advance, thus data assimilation can handle parameter uncertainty

more efficiently, leading to the best results among the five. Since IC-ObsInt and IC-Flux are created by

adding observation information or simple infiltration information, they perform better than that with IC-

HfSatu but worse than warm-up methods. Similarly, the assimilation results for IES with IC-WUP and

IC-WUE are also the best, while those with IC-HfSatu have the worst parameter estimation in the five

initialization methods (Fig. 6(b)). Moreover, the parameter estimations with IC-WUP and IC-WUE

require much fewer iteration steps (at about 7[th] iterations) than the other methods. In addition, by

comparing Figs. 6(a) and 6(b), the cases using IES shows superior results than those using EnKF,

indicating a better ability for IES to estimate individual parameter in variably saturated model. However,

since IES estimates parameter iteratively, it has a much larger computational cost than EnKF when using

warm up methods.

An important issue in data assimilation based on Monte-Carlo method is filter inbreeding (Hendricks

Franssen and Kinzelbach, 2008), which underestimates the ensemble variance over time/iteration, which

can lead to poor performance of parameter updating. To determine if our data assimilation runs   are

affected by filter inbreeding, the temporal change of the standard deviation of estimated $\ln K_s$ are plotted

in Figs. 6(c) and 6(d). In general, the standard deviation of estimated $\ln K_s$ decline gradually with

assimilation steps (EnKF) or iteration steps (ES), which agree well with the differences between the

estimates and the true values (Figs. 6(a) and 6(b)). Thus, filter inbreeding problem is not significant here,

and the data assimilation results are reliable.

To evaluate the effects of UIC in multi-parameter inverse problem, the $RMSE_m$ results of $K_s$, $\alpha$, and

$n$ estimates of Case 3 and Case 4 are presented in Fig. 7. In general, the $RMSE_m$ results of $n$ are the smallest

no matter using EnKF or IES, while the $RMSE_m$ of $\alpha$ is the largest. A cross-correlation analysis indicates

that soil moisture observations are very insensitive to parameter $\alpha$ with a free drainage boundary condition,

which agree with the results of Hu et al., (2017). In Fig. 7(a), similar to the conclusion of one-parameter

inverse problem, the parameter estimation results of $K_s$ and $\alpha$ with IC-HfSatu and IC-ObsInt are worse

than those of IC-WUP and IC-WUE. There is not much difference between the $n$ estimates under various

initial conditions, implying that $n$ is less affected by UIC when estimating $K_s$, $\alpha$ and $n$ simultaneously.

Compared with EnKF, IES shows a smaller $RMSE_m$ (Fig. 7(b)) of below 5 for all parameters, indicating

IES can also perform better in multi-parameter inverse problem. However, the assimilation results with

various initialization methods do not show much difference, implying that the final $RMSE_m$ values are not

significantly affected by UIC, possibly due to abundant observations available over one year. However,

long-term observation data may not be available in many cases.

To examine the impact of assimilation time on parameter estimation with IES, a case with shorter

assimilation period (60 days) is conducted. Fig. 8 shows the $RMSE_m$ results and it is inferior to than in

Case 4, where the simulation time is one year (Fig. 8(b)). Nevertheless, the effects of assimilation time

on parameter estimation are different for different parameters. For instance, parameter $n$ can still be

estimated quite well in this case. In addition, though the assimilation results of $K_s$ degraded with a 60-



days simulation, the variation of $K_s$ estimation values among different initialization methods is small. The number of observation can greatly affect the estimation of parameter $\alpha$, since $RMSE_m$ of $\alpha$ in Case 5 (25.64, 35.06, 8.52, 5.76 and 5.76) is much larger than those in Case 4 (1.19, 2.12, 4.19, 2.81 and 2.39).

## 4. Filed validation

In order to examine the applicability of the conclusions drawn from synthetic case, a field experiment is conducted in the irrigation-drainage experimental site of Wuhan University (Li et al., 2018) (Fig. 9(a)). Meteorological data, including air temperature, relative humidity, atmospheric pressure, incident solar radiation, and precipitation, is continuously monitored by an automatic weather station (LoggerNet 4.0), which can be used as upper boundary condition after the calculation of the potential evaporation on the

bare soil (see Fig. 10(a)). A vertically-inserted frequency domain reflectometry (FDR) tube was used to monitor soil moisture (Fig. 9(b)). The in-situ soil moisture observation was measured every 3 days. The tube gave soil moisture measurements at the depths of 10, 20 and 30 cm. During 18[th] April 2017 to 30[th] May 2017, the measurements were repeated 14 times and 42 soil moisture data were collected (see Fig. 10(b)). Besides, the soil moisture at the depth of 10 cm, 20 cm, 30 cm, 40 cm, 60 cm and 80 cm at the

beginning of the simulation time is also available to construct an initial profile via interpolation with IC-ObsInt.

### 4.1 General description of the experimental case

To analyze the experimental data, the 1-D numerical domain is set as 2 m and discretized in 50 grids. The top 40 grids have a size of 2.5 cm and the rest has a size of 10 cm. The upper boundary is set as an

atmospheric boundary using the data shown in Fig 10(a) and the bottom boundary is set to be free drainage since the groundwater table is much deeper than the bottom of the domain.

The prior parameter distributions follows the study of Li et al. (2018). The saturated soil moisture $\theta_s$ and residual soil moisture $\theta_r$ are given as 0.43 and 0.078, while the other hydraulic parameters are to be estimated. The initial means of $K_s$, $\alpha$ and $n$ are set as 1 md$^{-1}$, 5 m$^{-1}$ and 2, and the initial variances of

the natural logarithm of them are set as 0.22, 0.16 and 0.003. The data from 18th April through 27th April are used for calibration, while the remaining data are reserved for validation.

In addition, according to our prior knowledge, the meteorology of Wuhan is semi-arid conditions and the soil of experimental site is sandy loam. We use a warm-up time of 242 days based on our investigation





in Section 3.2.2.

## 4.2 Results

The assimilation results with four different initialization results (IC-HfSatu, IC-ObsInt, IC-Flux and IC-WUP) are presented in this part.Since the true hydraulic parameters at the experimental site are unknown, we assess the estimation by comparing the predicted and observed soil moistures during the validation period. The $RMSE_{obs}$ for soil moisture predictions under different assimilation scenarios are listed in Table 4. Generally speaking, $RMSE_{obs}$ with IC-WUP are again the smallest, while IC-HfSatu has the largest $RMSE_{obs}$ values.

In order to evaluate the overall performances of four initialization methods, the soil moisture observations and predictions at all depths are plotted in Fig. 11. The coefficients of determination under the four scenarios are 0.033, 0.599, 0.083 and 0.553, and the $RMSE_{obs}$ are 0.045, 0.037, 0.036, 0.028 respectively. As shown in Fig. 11(a) and Fig. 11(c), IC-HfSatu and IC-Flux show very large scattering, indicating a bad prediction performance. A significant improvement is found in IC-WUP with a large $R^2$ and the smallest $RMSE_{obs}$ value, as shown in Fig. 11(d). Surprisingly, IC-ObsInt has the largest $R^2$ among four in this case, though its $RMSE_{obs}$ value is bigger than that of IC-WUP. The simulation of real-world problems may have uncertainties that are considered in data assimilation. For instance, the meteorological data prior to the simulation for warming up may not be accurate. IC-ObsInt, on the other hand, consistently utilize the soil moisture observations for both initialization and predictions. It should be noted that IC-ObsInt may not be applicable in the case when soil moisture profile has large variation, e.g., discontinuous soil moisture in layered soils. In summary, as also demonstrated in the numerical case studies (Section 3.3), the model with initial condition using the warm-up method results in low uncertainty and superior soil moisture predictions even if the calibration data are insufficient.

## 5. Discussion and Conclusions

The study investigates the effects of UIC on variably saturated flow simulations subject to different soil hydraulic parameters, meteorological conditions and soil profile lengths. Two common approaches (Spin-up and Monte-Carlo methods) are applied to explore the required warm-up time $t_{wu}$ and temporal behavior of UIC. In addition, the data assimilation performances with five common initialization approaches are compared using synthetic experiments and a field soil moisture dataset.




Under atmospheric condition, the soil moisture value near the upper boundary could approach its upper and lower bounds (saturated water content and residual water content) due to rainfall and evaporation. This will significantly reduce the UIC of soil moisture profile near the soil surface. Moreover,

the wetter (drier) soil drains more (less) water and evaporates more (less) water, making UIC of soil moisture dissipates rapidly. Our investigation of the influence of soil texture and boundary condition on UIC shows, as expected, the coarse-textured soil results in faster reduction of soil moisture UIC because of fast redistribution of water in sandy soil. Regarding the influence of boundary conditions, we found heavy rainfall and long-term evaporation can reduce UIC significantly. The conclusion agrees with the

conclusions reported by Castillo et al., (2003) and Seck et al., (2014). Although $t_{wu}$ for sandy soil is very small, they could be very large for other soils (less than one day versus more than 10 years in Fig. 4). The length of soil profile plays an important role in UIC since UIC decays from the boundaries. As a result, the UIC could exist persistently in a very thick vadose zone. Our findings imply that UIC dissipation depends nonlinearly on soil type, meteorological condition, and soil profile lengths—care should be taken

when handling UIC in vadose zone modeling.

Ideally, the initial ensemble should represent the error statistics of the initial guess for the model state during data assimilation (Evensen, 2003). Thus, effort should be invested to reduce the impact of UIC on data assimilation. Methods which do not consider the UIC (i.e., assuming an initial ensemble without uncertainty, which was used in some studies, e.g., Shi et al., 2015) can induce significant bias according

to our data assimilation results. By constructing initial condition using the information of observations or boundary condition (averaged flux), the data assimilation results can be improved. However, these two initialization methods are also suboptimal, due to simplification of the complex initial condition. By warming up model with available meteorological data, the initialization methods can improve data assimilation results.

Our work leads to the following major conclusions:

1. Spin-up method and Monte-Carlo method can both quantify UIC and they agree well with each other after a sufficiently long simulation. A threshold of 0.5% for percentage cutoff $PC$ or ensemble spread $S_p$ is recommended to determine the warm-up time.

2. Warm-up time varies nonlinearly with soil textures, meteorological conditions, and soil profile



length.

3. IES shows better performance than EnKF in the strongly non-linear problem and is affected less by the UIC if warm-up method is implemented at the beginning of the simulation for every iteration. For both algorithms, the estimation of $\alpha$ is the most difficult while the parameter $n$ can be estimated more easily in the multi-parameter inverse problem.

4. The following procedure is recommended to initialize soil water model if meteorological data are available: 1) Evaluate the approximate warm-up time based on the model settings; 2) Initialize the model using the method of WUP and make sure the warming up time is larger than the required $t_{wu}$; 3) Run the simulation with the initial condition obtained in step 2. If the preceding meteorological data are not available, WUE is an alternative to obtain initial condition. ObsInt is also a practical choice if dense soil

moisture observations at the beginning of simulation are available.

Further research may examine the performance of these initialization methods in three-dimensional flow conditions and for large-scale problems. Our approach can be applied to models with multiple soil layers for the parameter estimation and to identify the warm-up time.

*Data/code availability.* All the data used in this study can be requested by email to the corresponding

author Yuanyuan Zha at zhayuan87@gmail.com.

*Author contribution*: Danyang Yu, Yuanyuan Zha and Jinzhong Yang developed the new package for soil water movement based on a switching Richards' equation; Danyang Yu and Yuanyuan Zha developed the data assimilation methods of EnKF and IES. Seven of the co-authors made non-negligible efforts preparing the manuscript.

*Competing interests*: The authors declare that they have no conflict of interest.

Acknowledgements. This work is supported by Natural Science Foundation of China through grants No. 51609173, 51779179, 51479144, 51522904. The authors appreciate Michael Tso (Lancaster University, U.K.) for editing the manuscript.






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



**Table 1.** The candidates of initialization scheme and default model settings.

| Parameter definition | Value |
|---|---|
| Initial condition | a uniform 50% relative saturation over the soil profile (IC-HfSatu) |
| | a linear interpolation between observations (IC-ObsInt) |
| | a steady soil moisture profile induced with flux=1 mm/d (IC-Flux) |
| | warming up with preceding meteorological data (IC-WUP) |
| | warming up with experiment-period meteorological data (IC-WUE) |
| Number of soil layers | 1 |
| Thickness of soil zone | 3 m |
| Soil hydraulic properties | Loam |
| Upper boundary | M-SC |
| Bottom boundary | Free drainage |
| Number of grids | 60 (with the size of 5 cm) |






**Table 2.** Soil hydraulic parameters used in simulation.

| Soil | $\theta_s$ | $\theta_r$ | $K_s$/md$^{-1}$ | $\alpha$/m$^{-1}$ | $n$ |
|------|------|------|------|------|------|
| Sand | 0.43 | 0.045 | 7.128 | 14.5 | 2.68 |
| Loam | 0.43 | 0.078 | 0.2496 | 3.6 | 1.56 |
| Silt | 0.46 | 0.034 | 0.06 | 1.6 | 1.37 |
| Clay loam | 0.41 | 0.095 | 0.062 | 1.9 | 1.31 |



**Table 3.** Case summary for parameter estimation within EnKF and IES.

| Case | Framework | Simulation time | Description |
| --- | --- | --- | --- |
| Case 1 | EnKF | | |
| | | One year | Individual parameter estimation; |
| Case 2 | IES | | |
| Case 3 | EnKF | | |
| | | One year | Multi-parameter estimation |
| Case 4 | IES | | |
| Case 5 | IES | 60 days | Multi-parameter estimation |





**Table 4.** *RMSE$_{obs}$* results for the soil moisture predictions at observation points with different initial conditions in

695                                    the experimental case.

| Initial condition | 10cm | 20cm | 30cm |
|---|---|---|---|
| IC-HfSatu | 0.0232 | 0.0271 | 0.0280 |
| IC-ObsInt | 0.0286 | 0.0187 | 0.0134 |
| IC-Flux | 0.0198 | 0.0222 | 0.0206 |
| IC-WUP | 0.0180 | 0.0153 | 0.0155 |

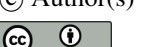



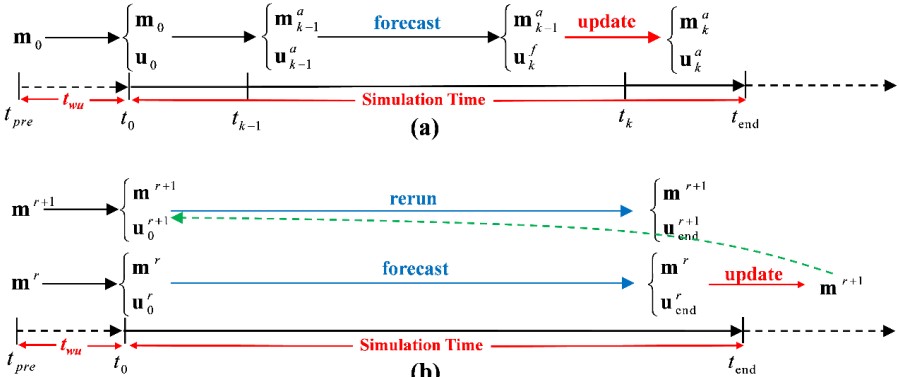

**Fig. 1.** Flowcharts of simulation period (or data assimilation period with (a) ensemble Kalman filter (EnKF) and (b) iterative ensemble smoother (IES)) and warming up period. $t_0$ is the initial time and $t_{end}$ is the end time of simulation. $\mathbf{m}_k$ and $\mathbf{u}_k$ are the vectors of model parameters (e.g., hydraulic conductivity) and state variables (e.g., soil moisture), respectively, at time $t_k$, while $\mathbf{m}^r$ and $\mathbf{u}^r$ are the vectors at iteration $r$; the superscripts $a$ and $f$ refer to model analysis and forecast (or initial guess). Besides, the period between $t_{pre}$ and $t_0$ donates the process of warming up, and $t_{wu}$ is the required warm-up time.





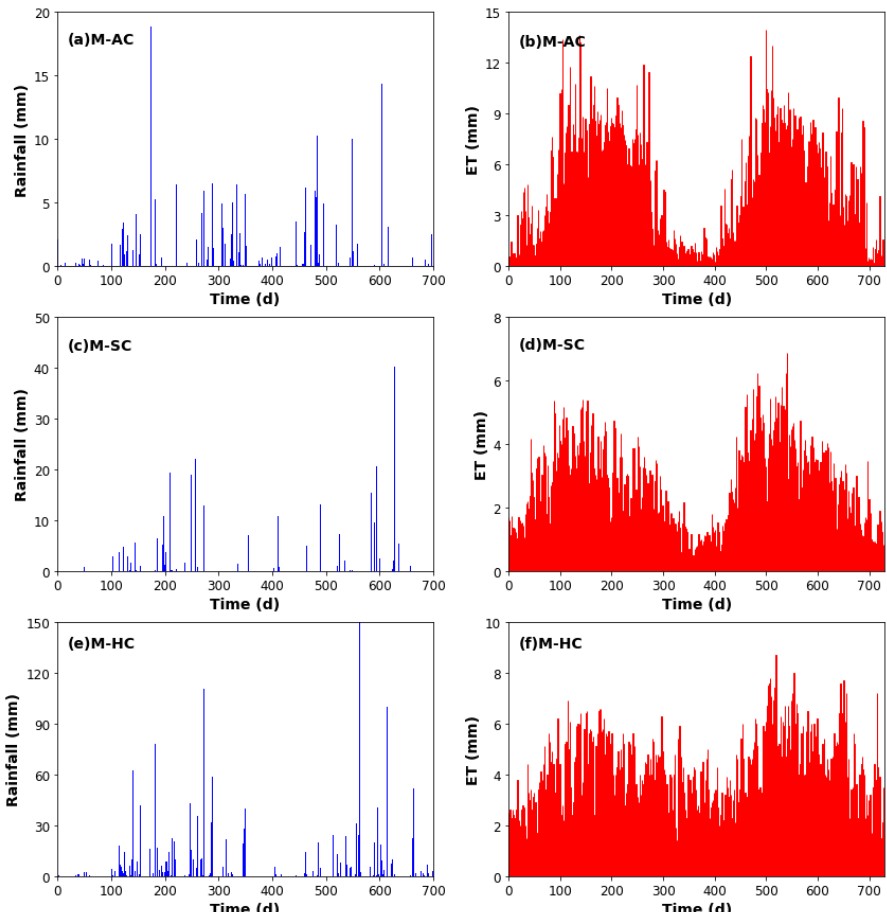


**Fig. 2.** Synthetic rainfall (blue bars) and potential evaporation (red bars) of three typical climates: (a) synthetic rainfall of arid climate, (b) synthetic potential evaporation of arid climate, (c) synthetic rainfall of semi-arid climate, (d) synthetic potential evaporation of semi-arid climate, (e) synthetic rainfall of humid climate, (f) synthetic potential evaporation of humid climate.






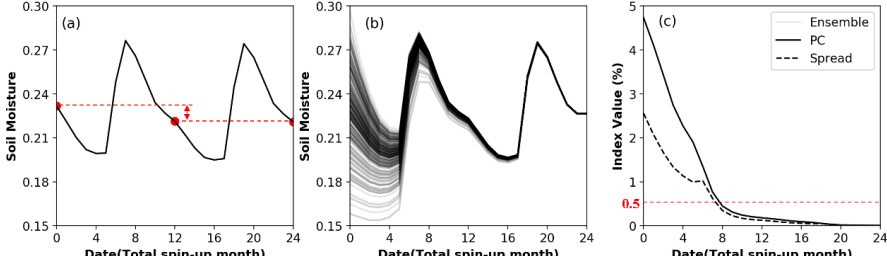

**Fig. 3.** Comparison of Spin-up and Monte-Carlo methods in determining warm-up time. (a) Spin-up method with monthly-averaged soil moisture versus time by running a simulation recursively for several years, (b) Monte-Carlo method with monthly-averaged soil moisture of different realizations versus time based on various initial conditions, and (c) Comparison of PC and $S_p$ versus time. For the purpose of demonstration, the parameter uncertainty is not considered.



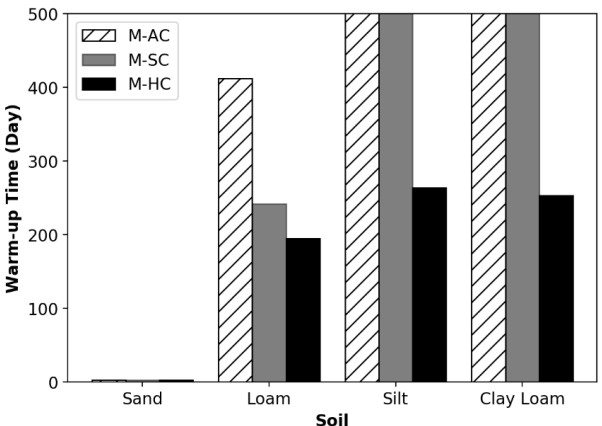

**Fig. 4.** The length of warm-up time $t_{wu}$ with various soils and meteorological conditions. Note that $t_{wu}$ of Silt and

Clay loam with M-AC and M-SC exceed 10 years, and they are trimmed for visualization purpose.





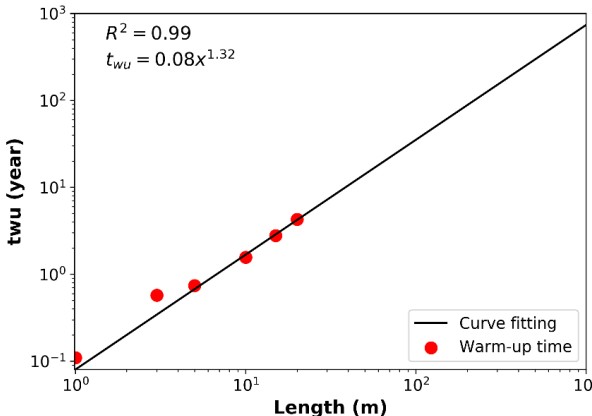

**Fig. 5.** The relationship between the length of soil profile and warm-up time $t_{wu}$.



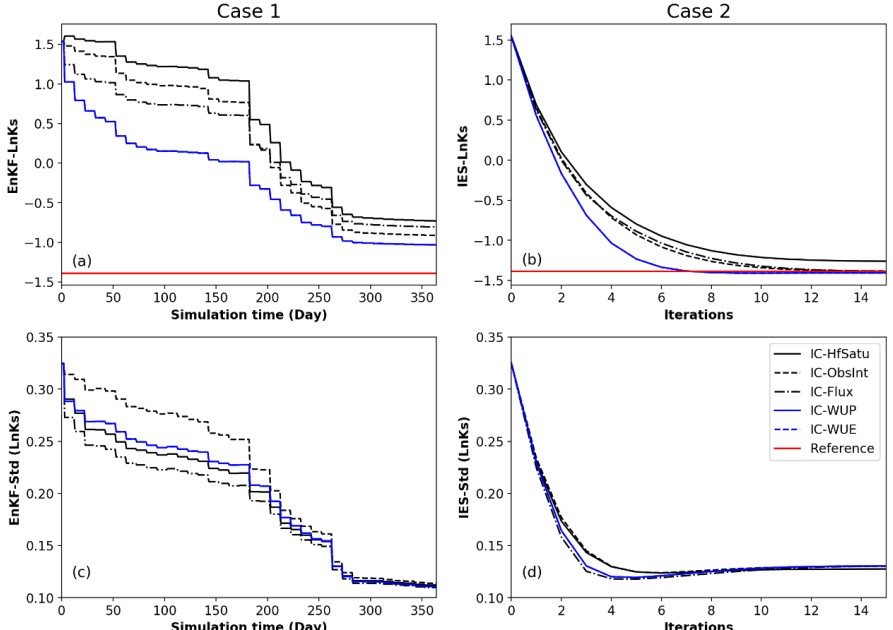


**Fig. 6.** The results of ln$K_s$ estimations (first row) and their associated standard deviations (second row) within two data assimilation frameworks (left: EnKF; right: IES) under five initialization methods.





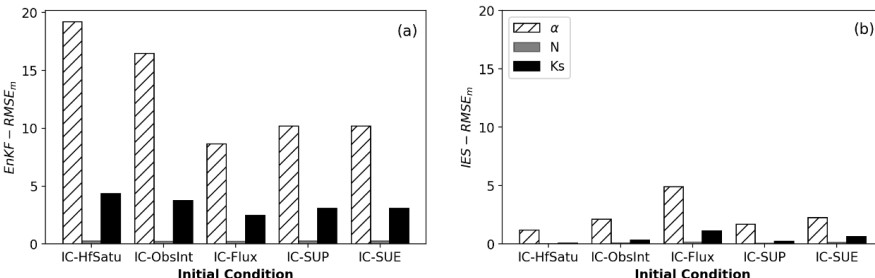

**Fig. 7.** The $RMSE_m$ results of parameter estimations ($\alpha$, $n$ and $K_s$) under five initialization methods with (a) EnKF and (b) IES.





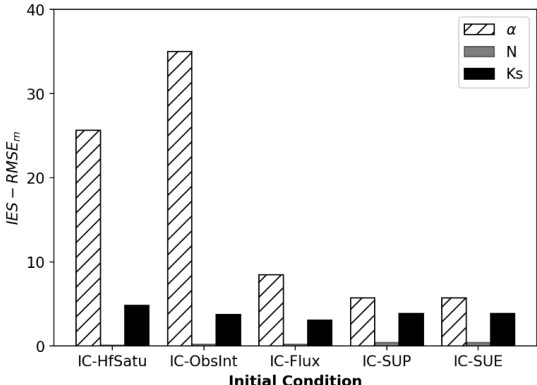

**Fig. 8.** The $RMSE_m$ results of parameter estimations under five initialization methods with IES when the

simulation time is 60 days.





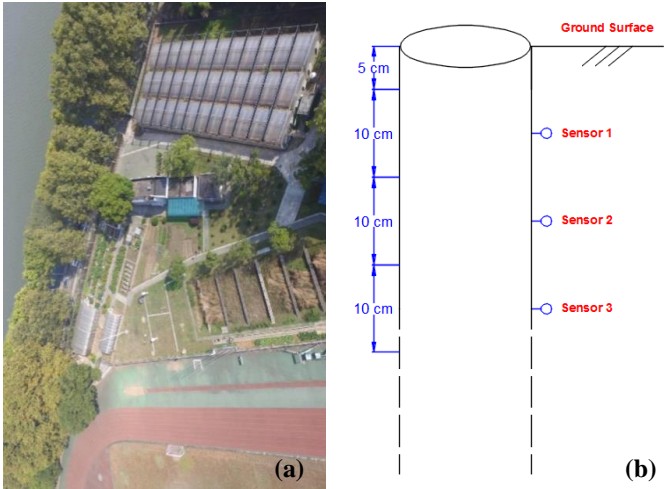

**Fig. 9.** The experimental site: (a) plan view, and (b) the cross-sectional view of the FDR sensor.





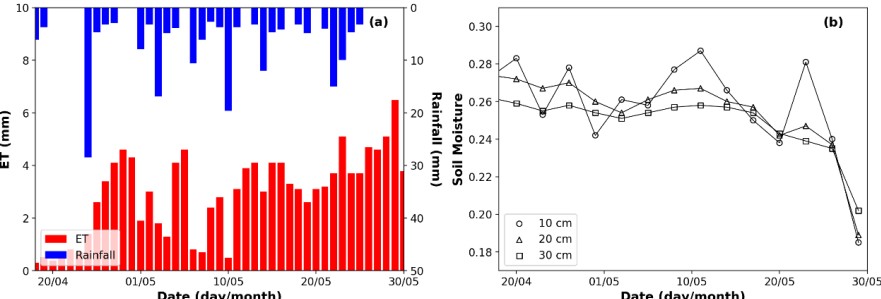


**Fig. 10.** The meteorological information and observed soil moistures over the experimental time. (a) Observed rainfall and calculated potential evaporation. (b) Temporal change of soil moisture data at three different observed depths (10 cm, 20 cm and 30 cm).





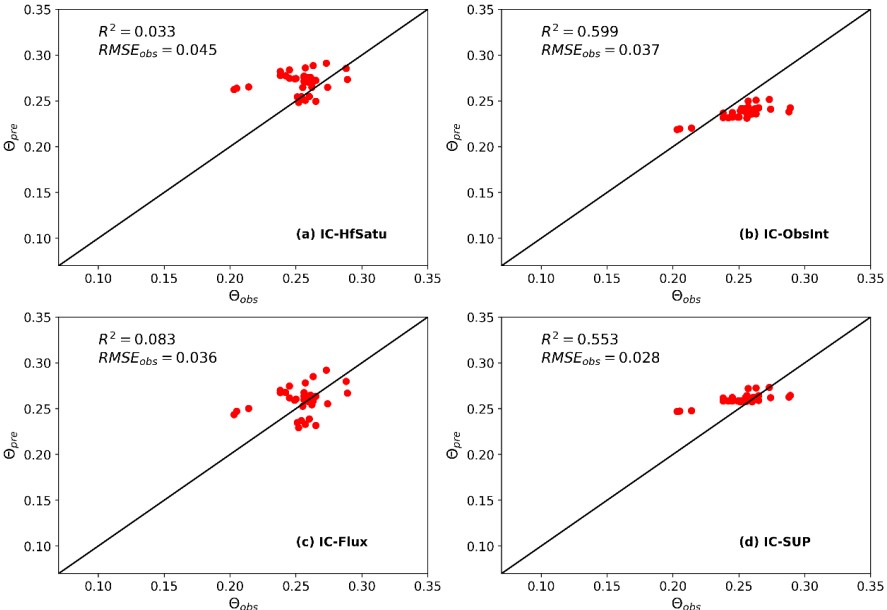


**Fig. 11.** The comparisons between soil moisture observations and predictions at all depths with four different initial

conditions.