# Peer review of "On the uncertainty of initial condition and initialization approaches in variably saturated flow modeling"

_Hydrology and Earth System Sciences, 2018_

## Referee Comment (RC1) · Anonymous Referee #1 · 26 Jan 2019

The manuscript investigates the effect of the uncertainty of the initial conditions in the context of soil water movement described by the Richards equation. First the necessary warm-up times for different soils and climates are determined and then the effects of different methods to describe the initial condition on a subsequent data assimilation are compared. The comparison is additionally shown on a real-world case.

I think the manuscript is interesting and and shows the effects of the uncertain initial conditions nicely. I have few comments that may require some additional investigation or discussion. However, the manuscript is sometimes difficult to read and could be clearer. Therefore, many comments ask for some clarification.

**Comparison of the different initial conditions:**

[Figure]

General comment:

The required computational power varies between the different initial conditions. The most expensive ones (with warm-up period) seem to give the best results in the subsequent data assimilation. I would find it very interesting whether this finding holds if for each method a similar total computation time (computation time for intitial condition + computation time for data assimilation) is available. This means that e.g. IC-ObsInt or IC-Flux could afford more ensemble members than IC-WUP. The question is then, if e.g a higher number of ensemble members (in combination with a larger uncertainty in the representation of the initial condition) of IC-ObsInt or IC-Flux could lead to similar, or even better results.

Specific comments:

Line 222-226: When the initial condition ensembles are generated for IC-HfSatu, IC-ObsInt and IC-Flux, is uncertainty added? How exactly? The uncertainty in the initial water content must be represented in the initial ensemble. If no uncertainty is added, this could explain partly the inferior result compared to IC-WUP. Please clarify and discuss.

Line 247: The spatial resolution of 5cm is rather low for a 1 dimensional case. Is this a computational limitation? Otherwise, I would suggest to reduce the grid size to e.g. 2cm. This is especially relevant for sandy soils where sharp infiltration fronts can develop and require such high resolutions. Could this impact the results?

Lines 379-383 and Figure 7: The biases of Ks, $\alpha$, and n as well as their uncertainties differ. Therefore, their RMSEs should not be compared directly. I think it is not a meaningful result that $\alpha$, which has the largest initial bias and uncertainty, also has the larger RMSE and that n, which has the smallest initial bias and uncertainty, has the

smaller RMSE. Their relative improvement might be a better measure.

**Warm-up time:**

Specific comments:

Line 286: You find that $t_{wu}$ is less than one day for sand. I think that this result might be due to the chosen initial condition for the warm-up. It is true, that for the chosen high water contents sand will drain very fast and rapidly approach a similar water content state. However, in case of an initial condition in a very dry state (which should be relevant for the arid climate), the hydraulic conductivity of sand drops to very low values and the initial spread can extend for a very long time, or until a sufficiently large rain event increases the water content and then leads again to the rapid approach of the similar water content.

I think it would be interesting to investigate this by choosing a different (dry) initial condition. At least this should be discussed in the manuscript.

Line 273-275 and 477-478: Since you recommend the choice 0.5% as a threshold: Please explain why. What is the advantage? Why should I not choose the other mentioned thresholds (e.g. 1% or 0.1%)?

Lines 306-313 and Figure 5: If I understand correctly, you investigate when the uncertainty for the full profile drops below the 0.5% threshold. In addition (or possibly as replacement) I would find it interesting to see the spatially resolved times for each depth for the deepest profile (20m).

Line 152 (Equation 8): If the monthly average of the previous year is required, wouldn't that imply that PC is not defined for the first year? However Fig. 3 shows PC starting

from time 0. Please clarify.

Figure 3: Why does the water content state after 24 months differ between panel (a) and (b)? Since both are initialised with the same parameter values and the UIC has already decayed, they should show the same soil moisture. Please clarify.

**Details of the used data assimilation:**

Specific comments:

Lines 370-377: Based on Figure 6, I disagree with the statement that filter inbreeding is not a significant issue for the EnKF case. In Figure 6, it seems that the final parameter value does not change any more over time and is over 5 standard deviations away from the truth. This means that the uncertainty is too small. Part of the reason could be that the initial uncertainty is chosen way too small. It is over 9 standard deviations away from the true value. This makes it very difficult for the EnKF to find the true value. I would suggest to repeat the simulations with a larger parameter uncertainty.

Line 177-178: ". . ., uk are state variables (i.e., pressure head and soil moisture) . . .". Do you update water content and matric potential of the same node simultaneously in the augmented state? Due to their nonlinear relation, the analysis would lead to inconsistencies between water content and matric potential for the analysis. How is this handled in the forecast? Please clarify.

**Technical comments:**

Lines 228-238: This part describes IC-WUP and IC-WUE. However, this is not a general description. In Section 3.2, when the spin up periods are investigated, a different

procedure is used. This confused me when reading the paper the first time. Please, only mention the general settings in 3.1 (i.e. climates, soils and model representation), and not specifics that only apply to 3.2 or 3.3. Therefore I would suggest to move this part to Section 3.3.

Additionally, here it is not clear how the parameter and initial condition ensembles are exactly generated. Please clarify.

Line 222-226: I think this part should be moved to Section 3.3 as well.

Line 243: "Fig. 1" should be "Fig. 2".

Line 254 and Fig. 3: The text mentions a simulation length of 10 years, the figure shows only 2 years. I would suggest to mention that you only show the first 2 years.

Lines 338-342: How many observations are there? In what depths are the observations? What is the assimilation frequency? Or is only a single observation in the depth of 10 cm assimilated every 10 days? If that is the case this has to be clarified.

Lines 343-350. I think this part should be moved to methods in Section 2.

Line 352-353 and Figure 6: I would mention that this is case 1 and case 2.

Line 399: "Field" instead of "Filed".

Figure 4: Since essentially the times for sand for all climates as well as silt and clay

loam for the M-Ac and the M-SC climate can not be properly displayed: Maybe the logarithm of the time could be more meaningful (like in Fig. 5).

Figure 6: The line for IC-WUE is essentially not visible. Is it below IC-WUP? At least mention this in the caption.

---

## Referee Comment (RC2) · Anonymous Referee #2 · 27 Jan 2019

This study investigates the temporal change of the uncertainty of initial condition in variably saturated flow model and assesses the impacts of several commonly-used initializing methods on results within various data assimilation frameworks. The topic is interesting and relevant to the topics of the Hydrology and Earth System Sciences. The manuscript is well-organized and easy to understand, although some of language, may be further refined and improved. The results and discussion are adequate to reach very instructive conclusions for variably saturated flow modeling. Several highlights for this manuscript: compared to previous researches on UIC issue, this study focuses on soil water modeling and makes a comparison between Monte Carlo (preferred by groundwater hydrologist) and Spinning up methods (preferred by surface water hydrologist). The investigation of warm-up time with different soil textures and depths is quite

interesting. The study of UIC propagation with data-model interaction is another merit. Therefore, I recommend this paper for publication in the Hydrology and Earth System Sciences, with a few comments. Major comments: 1) Authors have compared the difference of model outputs with various data assimilation framework (i.e., EnKF and IES). As the authors correctly point out, the ensemble size is an important factor for these two algorithms, which need to be discussed further. I encourage the authors to explore the effects of ensemble size on EnKF and IES with multiple test so that a suitable ensemble size for these two assimilation framework can be determined. 2) The synthetic case study present the proper warm-up time t[wu] versus different soil texture, soil depth, and meteorological conditions. While the relationship between t[wu] and meteorological conditions may be commonsense, the reveal of quantitative relationship between t[wu] and soil texture and soil depth is surprising and interesting, due to the fact that t[wu] changes abruptly from sand to finer texture, and it increases nonlinearly with the increase of soil depths. However, the soil is seldom homogeneous in natural conditions, especially for very long soil profile. The authors should at least present one simulating result of t[wu] for layered soil, which is more applicable for real-world case. I believe this should take too much work since it is one-dimensional model. Minor comments: Line 12: various initial condition »> various initial conditions Line 16: model initializing »> model initializing methods Line 28: delete in Line 48: a space between approaches and comma Line 61: hereafter referred »> hereafter referred to Line 77: delete the last the Line 81: initial ensemble are »> initial ensemble is Line 83: Currenlty »> Currently Line 110: Richards's »> Richards' Lines 129-130: as state-dependent, atmospeheric boundary condition (try to be more concise here and some other statements) Line 135: detemined »> determined Line 141: use UIC instead Eqs. (9-10): try to use one equation instead and shorten the description of the equation. Line 172: assimilation + approach Line 210: which lamta values you use in the simulations? Line 222: perscribe »> prescribe Line 223: availablitity »> availability Line 256: be consistent using itatic or not for PC. Line 256: why 3%? Line 335: warms »> warm Line 356: delete both Line 358-359: thus »> and thus Line 372: multiple spaces

between runs and are Line 392: Change "than" to that Line 405: Which evapotranspiration model are you using? Line 427: needs a space after "part." Lines 443-444: "soil moisture profile has large variation, e.g., discontinuous soil moisture in layered soils." — it would be interesting to see an additional case for heterogeneous soils, and this also leads to another interesting question — what will happen if pressure head profile, which is continuous in heterogeneous soil, is used as initial condition. Please add some discussion on this topic. Line 452: atmospheric condition »> atmospheric boundary condition Conclusion 2: Please include more details and add quantitative conclusions for this. Errors in references: Line 566, Line 673, Line 610, Line 639.

---

## Referee Comment (RC3) · Anonymous Referee #3 · 26 Feb 2019

This paper studies an important problem of soil water modeling: the uncertainty of initial condition (UIC) through analyzing the effects of different initial conditions on parameter estimation within two data assimilation frameworks. I believe this work provides useful insights to improve our understanding of uncertainty of initial conditions. I would be in favor of publication after the authors addressed the comments given below. Comments: 1. The grammar of this paper needs some improvements, some small grammar errors can be found. 2. The quantification of initial condition uncertainty (UIC) is unclear, especially for the usage of data assimilation method. I don't follow how two methods combined. 3. The purposes of using data assimilation method and its relationships to results and conclusions are unclear. 4. Please be more specific about why using both experimental and field model, and how different their results are. 5. Please describe

more details about the novelty of this paper, it seems there is no new method involved, and I am not sure how useful and novel the conclusions are.

---

## Author Comment (AC1) · 9 Mar 2019

Reply to comments from Anonymous Referee #1

**General Comments**

The manuscript investigates the effect of the uncertainty of the initial conditions in the context of soil water movement described by the Richards equation. First the necessary warm-up times for different soils and climates are determined and then the effects of different methods to describe the initial condition on a subsequent data assimilation are compared. The comparison is additionally shown on a real-world case. I think the manuscript is interesting and shows the effects of the uncertain initial conditions nicely. I have few comments that may require some additional investigation or discussion. However, the manuscript is sometimes difficult to read and could be clearer. Therefore, many comments ask for some clarification.

**[Response]**

Thanks for your positive comments. We have improved the manuscript according to your suggestions.

**Major comments:**

1. The required computational power varies between the different initial conditions. The most expensive ones (with warm-up period) seem to give the best results in the subsequent data assimilation. I would find it very interesting whether this finding holds if for each method a similar total computation time (computation time for initial condition + computation time for data assimilation) is available. This means that e.g. IC-ObsInt or IC-Flux could afford more ensemble members than IC-WUP. The question is then, if e.g a higher number of ensemble members (in combination with a larger uncertainty in the representation of the initial condition) of IC-ObsInt or IC-Flux could lead to similar, or even better results.

**[Response]**

Thank you for your valuable suggestion. First of all, we must apologize that we have

used mistaken model input folders when analyzing the data from WUE and WUP, and two identical curves for WUE and WUP were generated. Thanks for raising this question, leading to the discovery of this mistake. Nevertheless, the general conclusions from this figure still hold. Several minor modifications are given below (please see our response to your last comment) and will be reflected in the revised manuscript.

Then, in order to investigate whether the results for each method holds if with a similar computation time, we have added the cases for  $\ln K_s$  estimations utilizing the initial condition IC-HfSatu and IC-ObsInt with the ensemble size of 500 (hereafter referred as IC-HfSatu-500 and IC-ObsInt-500) in the manuscript. The computational costs of them are similar with that of IC-WUP, IC-WUE and IC-Flux. According to the results, we added a new figure and corresponding discussion in the manuscript about the effects of ensemble size on parameter estimation.

Figure a1. The impacts of ensemble size and the uncertainty of initial ensemble on the results of  $\ln K_s$  estimations within EnKF and IES.

The effects of increasing ensemble size are totally different for EnKF and IES. In Figure a1(a), when the ensemble size grows to 500, the assimilation results of IC-HfSatu-500 are worse than that of IC-ObsInt-500, implying that the initialization methods still plays an important role in parameter revisions even with a larger number of ensemble size. Besides, compared with the IC-ObsInt and IC-HfSatu with the ensemble size of 300 (Fig. 6(a)), IC-ObsInt-500 and IC-HfSatu-500 both show better data assimilation results,

indicating that the results of parameter estimations can be improved by increasing ensemble size for EnKF. The results are reasonable since the cross-correlation between model parameters and states can be better calculated with a large number of realizations. With the similar computational cost, the UIC of IC-ObsInt-500 stabilize the update steps, making the results better than IC-WUP. On the contrary, the impacts of improving ensemble size are slight for IES. As plotted in Figure a1(b), the data assimilation results of IC-HfSatu-500 and IC-ObsInt-500 are similar with that of 300. Since IES is a kind of iterative history-matching algorithm, the nonlinear relationship can be well calculated during the iterations, while the UIC is existed in the whole simulation. Therefore, warm-up methods show better data assimilation results within IES.

2. Line 222-226: When the initial condition ensembles are generated for IC-HfSatu, IC-ObsInt and IC-Flux, is uncertainty added? How exactly? The uncertainty in the initial water content must be represented in the initial ensemble. If no uncertainty is added, this could explain partly the inferior result compared to IC-WUP. Please clarify and discuss.

**[Response]**

Thank you for your careful reading. We are sorry that we did not explain the problem clearly.

(1) The initial conditions of IC-HfSatu and IC-ObsInt were assumed to be deterministic without uncertainty in the last manuscript. In contrast, IC-Flux was conducted by warming up the model for a period (length = "warm-up" time  $t_{wu}$ ) with a constant infiltration flux until a steady-state soil profile can be obtained. Thus, the uncertainty of parameter is introduced to IC-Flux, IC-WUP and IC-WUE during the construction of initial ensembles.

(2) The IC-HfSatu and IC-ObsInt were assumed to be deterministic without uncertainty. They are the most common and convenient methods to initialize the soil water/hydrological model, while most applications of these two methods do not consider including the parameter uncertainty (much larger than the magnitude of observation error) during the construction of initial conditions. We want to know how exactly they affect the data assimilation results and whether we can utilize these simple methods to initialize the model within data assimilation framework.

(3) In order to explore the effects of the uncertainty of the initial ensemble on the parameter estimations, the standard deviation of initial ensemble from IC-WUP is counted (0.017). Then, a Gaussian noise (with a standard deviation of 0.017) is added to both IC-HfSatu and IC-ObsInt (hereafter referred as IC-HfSatu-Un and IC-ObsInt-Un). We compared the results of parameter estimations between them and the other initial conditions, as presented in the updated Figure a1.

**[Changes in the manuscript]**

(1) We will add the content "It should be noted that IC-HfSatu and IC-ObsInt are assumed to be deterministic without uncertainty, while for the IC-Flux, the steady state is constructed by warming up model for a period of time which introduces the uncertainty of parameters"

(2) We will add Figure a1 in the manuscript. As plotted in Figure a1, the data assimilation results of IC-HfSatu-Un and IC-ObsInt-Un are better than those of IC-HfSatu and IC-ObsInt (Fig. 6(b)), indicating that the results of parameter estimations can be improved by increasing the variance of initial ensemble for EnKF. That may because the covariance between parameters and states can be better calculated with a large uncertainty of initial ensemble. However, with respect to IES, the assimilation results of IC-HfSatu-Un and IC-ObsInt-Un are worse than those of IC-HfSatu and IC-ObsInt, implying that a large variance of initial ensemble may deteriorate the data assimilation results within IES, since this uncertainty is difficult to diminish during the iterations. Hence, the optimal initialization method for EnKF may be the ObsInt approach with a large ensemble size and suitable uncertainty, while for IES, warm-up method can be a better choice, since the uncertainty of initial condition and parameter can both decrease during the iterations.

3. Line 247: The spatial resolution of 5cm is rather low for a 1 dimensional case. Is this a computational limitation? Otherwise, I would suggest to reduce the grid size to e.g.

**2cm. This is especially relevant for sandy soils where sharp infiltration fronts can develop and require such high resolutions. Could this impact the results?**

**[Response]**

Thank you for your suggestion. To understand the effects of grid size on variably saturated flow, we compared the results of temporal change of the soil moisture profile for the sandy soil with the grid size of 2 cm and 5 cm, as presented in Figure below.

**Figure 2.** The temporal change of soil moisture profile for the sandy soil with the grid size of 2 cm (a) and 5 cm (b).

In general, the soil moisture values at various depths all drop significantly in a short time, since the sandy soil has a great ability of drainage. Besides, owing to the effects of meteorological condition, the soil moisture at the surface decreases more quickly than that at the deep layer. However, compared with the model outputs using the grid size of 2 cm (Figure 2(a)), the results with the grid size of 5 cm (Figure 2(b)) are extremely similar, showing a slight impact of grid size.

**Figure 3.** The spread index value for the sandy soil over time with the grid size of 2 cm and 5 cm.

In order to explore the effects of grid size on warm-up time, the spread value over time for the sandy soil is plotted in Figure 3. Generally speaking, the spread value with the grid size of 5 cm shows a similar trend with that under the grid size of 2 cm. Besides, although there is a deviation between two spread values after 2 days' simulation, the deviation is too small that can be ignored. Furthermore, the  $t_{wu}$  value of both cases are the same. Hence, the grid size has little effects on the soil moisture simulation and  $t_{wu}$ for the sandy soil in our case.

**[Changes in the manuscript]**

(*previous manuscript*) The flow domain is discretized into 60 grids with a grid size of 5 cm.

(*revised manuscript*) The flow domain is discretized into 60 grids with a grid size of 5 cm which has been proved to be sufficient for one-dimensional soil water flow modeling in our case (results not shown).

4. Lines 379-383 and Figure 7: The biases of Ks,  $\alpha$ , and n as well as their uncertainties differ. Therefore, their RMSEs should not be compared directly. I think it is not a meaningful result that  $\alpha$ , which has the largest initial bias and uncertainty, also has the larger RMSE and that n, which has the smallest initial bias and uncertainty, has the smaller RMSE. Their relative improvement might be a better measure.

**[Response]**

Thank you for your suggestion. As the reviewer pointed out, *RMSE* index may be not suitable to evaluate the data assimilation results of different parameters, since their uncertainties are different. To identify the improvement of data assimilation results, we will add the relative error index (RE) into the manuscript, which can be calculated as follow,

$$RE = \frac{RMSE_e}{RMSE_p} \tag{1}$$

where  $RMSE_e$  and  $RMSE_p$  represent the RMSE of the estimated and prior parameters. *RE* varies from 0 to positive infinity. As *RE* approaches to 0, the analysis results are considered to be identical, but a large value of *RE* (more than 1) indicates that the results of parameter estimation are inferior to the prior parameters. Compared with the *RMSE*, this index can better present the impacts of different types of parameters on the data assimilation results. Hence, we modified Fig. 7 and Fig. 8 in the manuscript.

[Changes in the manuscript]

---

## Author Comment (AC2) · 9 Mar 2019

Reply to comments from Anonymous Referee #2.

**General Comments**

*This study investigates the temporal change of the uncertainty of initial condition in variably saturated flow model and assesses the impacts of several commonly-used initializing methods on results within various data assimilation frameworks. The topic is interesting and relevant to the topics of the Hydrology and Earth System Sciences. The manuscript is well-organized and easy to understand, although some of language, may be further refined and improved. The results and discussion are adequate to reach very instructive conclusions for variably saturated flow modeling. Several highlights for this manuscript: compared to previous researches on UIC issue, this study focuses on soil water modeling and makes a comparison between Monte Carlo (preferred by groundwater hydrologist) and Spinning up methods (preferred by surface water hydrologist). The investigation of warm-up time with different soil textures and depths is quite interesting. The study of UIC propagation with data-model interaction is another merit. Therefore, I recommend this paper for publication in the Hydrology and Earth System Sciences, with a few comments.*

**[Response]**

Thank you for your positive comment!

**Major comments:**

*1) Authors have compared the difference of model outputs with various data assimilation framework (i.e., EnKF and IES). As the authors correctly point out, the ensemble size is an important factor for these two algorithms, which need to be discussed further. I encourage the authors to explore the effects of ensemble size on EnKF and IES with multiple test so that a suitable ensemble size for these two assimilation framework can be determined.*

**[Response]**

Thank you for your comment, according to the suggestions from you and another reviewer, we will add a new figure to explore the effects of ensemble size on the

parameter estimations within EnKF and IES. The impact of ensemble size on UIC seems different for various data assimilation framework.

**[Changes in the manuscript]**

[Figure]

**Figure 1.** The impacts of ensemble size and the uncertainty of initial ensemble on the results of $\ln K_s$ estimations within EnKF and IES.

The effects of increasing ensemble size are totally different for EnKF and IES. In Figure 1(a), when the ensemble size grows to 500, the assimilation results of IC-HfSatu-500 are worse than that of IC-ObsInt-500, implying that the initialization methods still plays an important role in parameter revisions even with a larger number of ensemble size. Besides, compared with the IC-ObsInt and IC-HfSatu with the ensemble size of 300 (Fig. 6(a)), IC-ObsInt-500 and IC-HfSatu-500 both show better data assimilation results, indicating that the results of parameter estimations can be improved by increasing ensemble size for EnKF. The results are reasonable since the cross-correlation between model parameters and states can be better calculated with a large number of realizations. With the similar computational cost, the UIC of IC-ObsInt-500 stabilize the update steps, making the results better than IC-WUP. On the contrary, the impacts of improving ensemble size are slight for IES. As plotted in Figure 1(b), the data assimilation results of IC-HfSatu-500 and IC-ObsInt-500 are similar with that of 300. Since IES is a kind of iterative history-matching algorithm, the nonlinear relationship can be well calculated during the iterations, while the UIC is existed in the whole simulation. Therefore, warm-up methods show better data assimilation results within IES.

*2) The synthetic case study present the proper warm-up time $t_{wu}$ versus different soil texture, soil depth, and meteorological conditions. While the relationship between $t_{wu}$ and meteorological conditions may be commonsense, the reveal of quantitative relationship between $t_{wu}$ and soil texture and soil depth is surprising and interesting, due to the fact that $t_{wu}$ changes abruptly from sand to finer texture, and it increases nonlinearly with the increase of soil depths. However, the soil is seldom homogeneous in natural conditions, especially for very long soil profile. The authors should at least present one simulating result of $t_{wu}$ for layered soil, which is more applicable for real-world case. I believe this should take too much work since it is one-dimensional model.*

**[Response]**

Thank you for your valuable comment. We have added a case to obtain the $t_{wu}$ for layered soil in our manuscript. The layered soil is consistent of four kinds of soil types, including loam (0 to 75 cm), clay loam (75 to 150 cm), silt (150 to 225 cm) and sand (225 to 300 cm).

**[Changes in the manuscript]**

[Figure]

**Fig. 4.** The length of warm-up time $t_{wu}$ with various soils and meteorological conditions. Note that $t_{wu}$ of Silt and Clay loam with M-AC and M-SC exceed 10 years as well as the $t_{wu}$ of multiple layers with M-AC. The consistent layers of heterogeneous soil are the loam (0-75 cm), clay loam (75-150 cm), silt (150-225 cm), and sand (225-300 cm).

In the revised manuscript, we expanded the results and demonstrate the difference of $t_{wu}$ value in the layered soil.

**Minor comments:**

*Line 12: various initial condition »> various initial conditions*

Thank you. Revised

*Line16: model initializing »> model initializing methods*

Thank you. This has been revised.

*Line 28: delete in*

Thank you. Revised.

*Line 48: a space between approaches and comma*

Thanks. The error has been corrected.

*Line 61: hereafter referred »> hereafter referred to*

Thank you. It is revised

*Line 77: delete the last the*

Thank you. It is modified.

*Line 81: initial ensemble are »> initial ensemble is*

Thank you. It is rewritten.

*Line83: Currenlty »> Currently*

Thank you. Revised

*Line 110: Richards's »> Richards'*

Thanks. Revised

*Lines 129-130: as state dependent, atmospherics boundary condition (try to be more concise here and some other statements)*

Thanks. It is rewritten.

*Line 135: detemined »> determined*

Thanks. Revised

*Line 141: use UIC instead Eqs. (9-10): try to use one equation instead and shorten the description of the equation.*

Thank you. It is revised.

*Line 172: assimilation + approach*

Thanks. Revised

*Line 210: which λ values you use in the simulations?*

Thanks. $\lambda$ is a dynamic stability multiplier during the iterations. The prior value of $\lambda$ is 10, but the value can be adjusted adaptably according to the data assimilation results at every iteration.

*Line 222: perscribe »> prescribe*

Thank you. It is revised.

*Line 223: availablitity »> availability*

Thanks. Revised.

*Line 256: be consistent using italic or not for PC.*

Thank you. This has been revised.

*Line 256: why 3%?*

Thank you. The Gaussian noise is determined as 3% according to the observation error of soil moisture since the uncertainty of parameter is not taken into consideration in this part. We have added a sentence to explain it.

*Line 335: warms »>warm*

Thanks. Revised.

*Line 356: delete both*

Thanks. This has been revised.

*Line 358-359: thus »> and thus*

Thanks. Revised.

*Line 372: multiple spaces between runs and are.*

Thanks. Revised.

*Change "than" to that*

Thanks. This has been revised.

*Line 405: Which evapotranspiration model are you using?*

Thank you for your comment. The potential evaporation is calculated by Penman-Monteith's equation. We will add an explanation in the manuscript.

*Line 427: needs a space after "part."*

Thanks. Revised.

*Lines 443-444: "soil moisture profile has large variation, e.g., discontinuous soil moisture in layered soils." — it would be interesting to see an additional case for heterogeneous soils, and this also leads to another interesting question — what will happen if pressure head profile, which is continuous in heterogeneous soil, is used as*

*initial condition. Please add some discussion on this topic.*

**[Response]**

Thank you for your valuable comments. We have added the case about $t_{wu}$ of multiple layers in the manuscript, please see Fig. 4 above. Regarding the topic about using initial pressure head as initial condition, we are going to discuss it from three aspects:

(1) It is easier to collect the soil moisture data than pressure head in vadose zone, so that we only use soil moisture as observation in this study. In heterogeneous soil, the pressure head profile is more continuous than the soil moisture profile, which may contribute to better model outputs.

(2) With respect to $t_{wu}$, the conversion relationship between the spread of soil moisture and pressure head is deterministic (i.e., the spread as well as the $t_{wu}$ value with pressure head profile can be derived from that with soil moisture profile). Thus, the effects of pressure head and soil moisture are slight on UIC.

(3) The impact of observation type (i.e., pressure head and soil moisture) on data assimilation results has been widely explored in previous studies (Shi et al. 2015), since they have various distributions, nonlinearity and observation errors. The deviation of data assimilation results between pressure head and soil moisture is less induced by UIC, so we are not going to discuss in detail here.

**[Changes in the manuscript]**

We will add some discussions about the effects of initial pressure head profile in heterogeneous soil in Section 5.

*Line 452: atmospheric condition »> atmospheric boundary condition*

Thanks. Revised.

*Conclusion 2: Please include more details and add quantitative conclusions for this.*

**[Response]**

Thank you. We have modified the conclusion 2 according to your suggestion.

**[Changes in the manuscript]**

Warm-up time varies nonlinearly with soil textures, meteorological conditions, and soil

profile. A very long time is needed to warm up the model for the fine-textured soil with an arid meteorological condition and a thick vadose zone.

*Errors in references: Line 566, Line 673, Line 610, Line 639.*

Thanks. Revised.

**Reference**

Shi, L., Song, X., Tong, J., Zhu, Y. and Zhang, Q.: Impacts of different types of measurements on estimating unsaturated flow parameters, J. Hydrol., 524, 549–561, doi:10.1016/j.jhydrol.2015.01.078, 2015.

---

## Author Comment (AC3) · 9 Mar 2019

Reply to comments from Anonymous Referee #3

**General Comments**

*This paper studies an important problem of soil water modeling: the uncertainty of initial condition (UIC) through analyzing the effects of different initial conditions on parameter estimation within two data assimilation frameworks. I believe this work provides useful insights to improve our understanding of uncertainty of initial conditions. I would be in favor of publication after the authors addressed the comments given below.*

**[Response]**

Thank you for your positive comment! We have revised our manuscript according to your suggestions.

**Comments:**

*1. The grammar of this paper needs some improvements, some small grammar errors can be found.*

Thank you for your suggestion. We have invited a native English speaker to proofread our manuscript.

*2. The quantification of initial condition uncertainty (UIC) is unclear, especially for the usage of data assimilation method. I don't follow how two methods combined.*

**[Response]**

Thank you for your valuable comment. We are sorry that we did not explain it clearly. In data assimilation problems, the parameter uncertainty is considered to be major uncertainty source, while the effects of initial condition are often ignored. In this study, we take the uncertainty of both into the consideration. In order to gain a better understanding of the propagation of the UIC, we firstly assumed all the hydraulic parameters (i.e., $K_s$, $\alpha$ and $n$) to be deterministic and the UIC is the only uncertainty source (Section 3.2). By utilizing the PC and Spread index, the effects of UIC over time

and the required warm-up time $t_{wu}$ (minimum time required for the model to warm up to eliminate the UIC) can be calculated. Then, we take both the uncertainty of UIC and parameter into the consideration and compared several common initialization methods for variably saturated model. And we found that when we warm up model more than $t_{wu}$, better data assimilation results can be obtained. Our work can be a reference for the data assimilation problem.

**[Changes in the manuscript]**

We have added some explanations and modified our manuscript to make the description clearer.

*3. The purposes of using data assimilation method and its relationships to results and conclusions are unclear.*

**[Response]**

Thank you for your comment.

(1) Accurate characterization of soil properties is essential for the precise predictions of water movement and solute distributions in the vadose zone. However, obtaining detailed knowledge of soil parameters is challenging for its difficulty in direct measurement. Yet, hydrological data, such as soil moisture and pressure head, can be collected more conveniently and provide additional information. Incorporating these observations into stochastic models helps to reduce the uncertainty of the parameters and prediction which is known as "data assimilation". In recent years, data assimilation has become a popular tool and been widely used in the field of vadose zone hydrology (Shi et al., 2015; Vereecken et al., 2010; Walker et al., 2001). Hence, it is necessary to fully investigate data assimilation problem. One of our purpose is to identify how exactly UIC affect the data assimilation results and propose a suitable initialization method for the model within various data assimilation framework.

(2) We drew two main conclusion about how UIC affects data assimilation results. First, by comparing sequential data assimilation (EnKF) and history-matching algorithm (IES), we found that the IES is affected less by the UIC if warm-up method is implemented at the beginning of the simulation for every iteration and with a long

period of observations, while regarding EnKF, data assimilation results can be improved by increasing ensemble size, but the effects of UIC should be paid more attention if a small ensemble size is employed. Second, by comparing data assimilation results, we propose a selection scheme for choosing a suitable approach of initializing variably saturated flow models within different data assimilation frameworks to minimize the influence of UIC.

**[Changes in the manuscript]**

We will improve our manuscript according to your suggestions. Some explanations will be added about our purposes of using data assimilation method and we will make its relationships to results and conclusions clearer.

*4. Please be more specific about why using both experimental and field model, and how different their results are.*

**[Response]**

Thank you for your comment. In this study, we used both synthetic and field observations. Synthetic data is generated by running the model, while the field data is collected in the experimental station. The true field observations may contain a lot of uncertainty such as artificial and natural error so that we cannot understand the reason which truly causes the change of observations. By utilizing the synthetic observations, we can obtain more accurate conclusions. Then, the field data can be used to validate these conclusions.

The conclusions are similar by using field or synthetic data, but the difference of results between various initialization methods are not so significant, since there are a lot of uncertainty in true observations.

**[Changes in the manuscript]**

We will add an explanation like "In order to examine the applicability of the conclusions drawn from synthetic case in the real-world, the true field observations are necessary to be incorporated into the model" in the manuscript.

*5. Please describe more details about the novelty of this paper, it seems there is no new*

*method involved, and I am not sure how useful and novel the conclusions are.*

**[Response]**

Thank you for your comment. Indeed, we did not have any new method involved. However, to the best of our knowledge, we are the first to systematically analyze the effects of initial conditions and initialization methods on the various data assimilation frameworks to date. The specific novelties includes three aspects.

(1) Two common approaches for quantifying the temporal evolution of initial condition uncertainty are compared.

(2) The influences of soil texture, meteorological condition and soil profile length on initial condition uncertainty evolution are exploited.

(3) Different approaches to initialize unsaturated-saturated flow models within two data assimilation framework are assessed.

According to our results, Spin-up method and Monte-Carlo method can both quantify UIC and they agree well with each other after a sufficiently long simulation. And we recommend a threshold of 0.5% for percentage cutoff PC or ensemble spread $S_p$ to balance the computation cost and the effects of UIC. Moreover, the relationship between warm-up time for variably saturated flow modeling and the model settings (soil textures, meteorological conditions and soil profile length) are quantitatively identified. In addition, UIC shows different impacts for IES and EnKF, and we propose a "warm-up" period before assimilating data in order to obtain a better performance for parameter and state estimation.

In conclusion, our work can be a reference for other study to choose a suitable approach of initializing variably saturated flow model within data assimilation framework to minimize the influence of UIC.

**[Changes in the manuscript]**

We will modify our manuscript according to the discussion above to make the novelties of the paper more apparent.

**Reference**

Ajami, H., McCabe, M. F., Evans, J. P. and Stisen, S.: Assessing the impact ofmodel

spin-up on surface water-groundwater interactions using an integrated hydrologic model, Water Resour. Res., 50, 1–21, doi:10.1002/2013WR014258.Received, 2014.

Brandhorst, N., Erdal, D. and Neuweiler, I.: Soil moisture prediction with the ensemble Kalman filter: Handling uncertainty of soil hydraulic parameters, Adv. Water Resour., 110(August), 360–370, doi:10.1016/j.advwatres.2017.10.022, 2017.

Li, C. and Ren, L.: Estimation of Unsaturated Soil Hydraulic Parameters Using the Ensemble Kalman Filter, Vadose Zo. J., 10(4), 1205, doi:10.2136/vzj2010.0159, 2011.

Montzka, C., Moradkhani, H., Weihermüller, L., Franssen, H. J. H., Canty, M. and Vereecken, H.: Hydraulic parameter estimation by remotely-sensed top soil moisture observations with the particle filter, J. Hydrol., 399(3–4), 410–421, doi:10.1016/j.jhydrol.2011.01.020, 2011.

Shi, L., Song, X., Tong, J., Zhu, Y. and Zhang, Q.: Impacts of different types of measurements on estimating unsaturated flow parameters, J. Hydrol., 524, 549–561, doi:10.1016/j.jhydrol.2015.01.078, 2015.

Vereecken, H., Huisman, J. A., Bogena, H., Vanderborght, J., Vrugt, J. A. and Hopmans, J. W.: On the value of soil moisture measurements in vadose zone hydrology: A review, Water Resour. Res., 46(4), 1–21, doi:10.1029/2008WR006829, 2010.

Walker, J. P., Willgoose, G. R. and Kalma, J. D.: One-Dimensional Soil Moisture Profile Retrieval by Assimilation of Near-Surface Measurements: A Simplified Soil Moisture Model and Field Application, J. Hydrometeorol., 2(4), 356–373, doi:10.1175/1525-7541(2001)002<0356:ODSMPR>2.0.CO;2, 2001.

Wu, C.-C. and Margulis, S. A.: Real-Time Soil Moisture and Salinity Profile Estimation Using Assimilation of Embedded Sensor Datastreams, Vadose Zo. J., 12(1), doi:10.2136/vzj2011.0176, 2013.

---

## Author Response (AR1)

Comments by the Associate Editor:

**Editor Decision**

*I received comments from three reviewers, two suggested major revision, and one (reviewer 2) suggested minor revision. All the three reviewers confirmed the contribution of the manuscript, but also pinpointed the problems of the manuscript. After received the comments, I carefully read the manuscript again and concur with the reviewers. Therefore, I suggested "reconsider after major revision". Please provide detailed replies to the comments made by the reviewers and revised your manuscript accordingly.*

**[Response]**

Thanks for handling our manuscript. We have improved our manuscript according to the referees' suggestions.

Reply to comments from Anonymous Referee #1

**General Comments**

*The manuscript investigates the effect of the uncertainty of the initial conditions in the context of soil water movement described by the Richards equation. First the necessary warm-up times for different soils and climates are determined and then the effects of different methods to describe the initial condition on a subsequent data assimilation are compared. The comparison is additionally shown on a real-world case. I think the manuscript is interesting and shows the effects of the uncertain initial conditions nicely. I have few comments that may require some additional investigation or discussion. However, the manuscript is sometimes difficult to read and could be clearer. Therefore, many comments ask for some clarification.*

**[Response]**

Thanks for your positive comments. We have improved the manuscript according to your suggestions.

**Major comments:**

*1. The required computational power varies between the different initial conditions. The most expensive ones (with warm-up period) seem to give the best results in the subsequent data assimilation. I would find it very interesting whether this finding holds if for each method a similar total computation time (computation time for initial condition + computation time for data assimilation) is available. This means that e.g. IC-ObsInt or IC-Flux could afford more ensemble members than IC-WUP. The question is then, if e.g a higher number of ensemble members (in combination with a larger uncertainty in the representation of the initial condition) of IC-ObsInt or IC-Flux could lead to similar, or even better results.*

**[Response]**

Thank you for your valuable suggestion. First of all, we must apologize that we have used mistaken model input folders when analyzing the data from WUE and WUP, and

two identical curves for WUE and WUP were generated. Thanks for raising this question, leading to the discovery of this mistake. Nevertheless, the general conclusions from this figure still hold. Several minor modifications are given below (please see our response to the last comment of yours) and will be reflected in the revised manuscript. Then, in order to investigate whether the conclusion for each method holds with a similar computational time and with a large uncertainty of initial state, we have added four cases for $\ln K_s$ estimations based on the initial condition IC-HfSatu and IC-ObsInt. In the first two cases, we increase the ensemble size of IC-HfSatu and IC-ObsInt to 500 (hereafter referred to as IC-HfSatu-500 and IC-ObsInt-500 in the manuscript) to explore the influences of ensemble size. The computational costs of them are similar to those of IC-WUP, IC-WUE and IC-Flux. Next, to further explore the effects of the uncertainty of the initial ensemble on the parameter estimations, we add a Gaussian noise (with a standard deviation of 0.017) to both IC-HfSatu-500 and IC-ObsInt-500 (hereafter referred to as IC-HfSatu-500-Un and IC-ObsInt-500-Un). The standard deviation of the Gaussian noise is calculated based on the spreading of IC-WUP initial ensemble, so that we can make sure the initial uncertainties of IC-HfSatu-500-Un and IC-ObsInt-500-Un are similar with IC-WUP.

According to the results of the four cases, we added a new figure (Fig. 7 in the revised manuscript) and corresponding discussion in the manuscript on the effects of ensemble size and the uncertainty of initial state on parameter estimation.

**[Changes in the manuscript]**

[Figure]

**Fig. 7. (revised manuscript)** The impacts of ensemble size and the uncertainty of initial state

on the results of $\ln K_s$ estimations using EnKF and IES.

The results of IC-HfSatu-500 and IC-ObsInt-500 with the ensemble size of 500 in Fig. 7 are similar with those of IC-HfSatu and IC-ObsInt (Fig. 6), indicating that the improvement of the parameter estimation result is slight when the ensemble size increases from 300 to 500. Hence, the ensemble size of 300 is sufficient for data assimilation problem in this study. In contrast, the influences of adding the uncertainty to the initial state on parameter estimation are totally different for EnKF and IES. Compared with the results of IC-ObsInt-500 and IC-HfSatu-500, the results of $\ln K_s$ estimation with IC-ObsInt-500-Un and IC-HfSatu-500-Un improve for EnKF (Fig. 7(a)), but deteriorate for IES (Fig. 7(b)). This may attribute to the diversity between two algorithms. EnKF is a sequential algorithm, so the state uncertainty introduced by UIC could decrease over assimilation steps. A larger ensemble state variance implemented at the beginning leads to a larger trust on data and thus a quicker update of parameter to truth, and can prevent EnKF from inbreeding, leading to a better result than that with initial condition of small variance. On the contrary, IES is a batch optimization method. The uncertainty of initial state exists at each iteration and has a negative effect on the model calibration during the whole simulation, worsening the parameter estimation results.

Moreover, the parameter estimation results of IC-WUP are still superior to those of IC-HfSatu-500-Un and IC-ObsInt-500-Un even they all have a similar computational cost, showing the promising performance of warm-up methods. The results are reasonable since all ensemble Kalman filter methods are affected by the quality of the auto-covariance matrix and the mean value of predicted state ensemble (Eqs. (12) and (13) for EnKF; Eqs. (15) and (16) for IES). For WUP method, the initial condition is constructed by warming up the model with the prior parameter, thus IC-WUP contains useful information of prior parameter, even it is biased. Besides, the state covariance matrix is implicitly inflated due to the introduction of uncertain prior parameter ensemble during warming up. These two aspects ensure the robust performance of warm-up methods. However, the initial state ensembles of IC-HfSatu-500-Un and IC-

ObsInt-500-Un are independent from the prior parameter, which introduces additional uncertainties, making the data assimilation results worse. Therefore, even using a larger ensemble size and enlarging the state uncertainty (covariance inflation), warm-up methods are still the optimal choice for both EnKF and IES algorithms.

*2. Line 222-226: When the initial condition ensembles are generated for IC-HfSatu, IC-ObsInt and IC-Flux, is uncertainty added? How exactly? The uncertainty in the initial water content must be represented in the initial ensemble. If no uncertainty is added, this could explain partly the inferior result compared to IC-WUP. Please clarify and discuss.*

**[Response]**

Thank you for your careful reading. We are sorry that we did not explain the problem clearly.

(1) The initial conditions of IC-HfSatu and IC-ObsInt were assumed to be deterministic without uncertainty in the original manuscript. In contrast, IC-Flux was conducted by warming up the model for a period (length = "warm-up" time $t_{wu}$) with a constant infiltration flux until a steady-state soil profile can be obtained. Thus, the uncertainty of parameter is introduced to IC-Flux, IC-WUP and IC-WUE during the construction of initial ensembles.

(2) IC-HfSatu and IC-ObsInt are the most common and convenient methods to initialize the soil water/hydrological model, while most applications of these two methods do not consider including the parameter uncertainty (much larger than the magnitude of observation error) during the construction of initial conditions. We want to know how exactly they affect the data assimilation results and whether we can utilize these first-cut methods to initialize the model within data assimilation framework.

(3) In order to further explore the effects of the uncertainty of the initial ensemble on the parameter estimations for these two approaches, we add a Gaussian noise (with a standard deviation of 0.017, which is calculated based on the spreading of IC-WUP initial ensemble) to both IC-HfSatu and IC-ObsInt (hereafter referred as IC-HfSatu-500-Un and IC-ObsInt-500-Un). We compared the results of parameter estimations

between them and the other initial conditions, as presented in the updated Fig. 7 (revised manuscript).

**[Changes in the manuscript]**

(1) We have added the content "In addition, IC-HfSatu and IC-ObsInt are assumed to be deterministic without uncertainty, while for the IC-Flux, IC-WUP and IC-WUE, the uncertainty of states are introduced by warming up the model with uncertain parameters." Please see lines 344-346.

(2) We have added the cases and discussions about the effects of the initial state with or without uncertainty. Please see our response to your first comment.

*3. Line 247: The spatial resolution of 5cm is rather low for a 1 dimensional case. Is this a computational limitation? Otherwise, I would suggest to reduce the grid size to e.g. 2cm. This is especially relevant for sandy soils where sharp infiltration fronts can develop and require such high resolutions. Could this impact the results?*

**[Response]**

Thank you for your suggestion. To understand the effects of grid size on the results, we compared the temporal change of the soil moisture profiles for the Loam soil (i.e., default soil type) with the grid size of 2 cm and 5 cm, as presented in Figure below.

[Figure]

**Figure 2.** The temporal change of soil moisture profile for the Loam soil with the grid size of 2 cm (a) and 5 cm (b).

We admitted that the grid probably is too coarse to accurately capture the soil moisture dynamics, especially the sharp wetting front. Nevertheless, the difference between the

overall results with the grid sizes of 2 cm (Figure 2(a)) and 5 cm (Figure 2(b)) is insignificant. Since our purpose is to explore the temporal change of UIC, which is an overall statistical index of the soil moisture profile, we think this grid is justifiable.

[Figure]

**Figure 3.** The spread index value for the Loam soil over time with the grid size of 2 cm and 5 cm.

In order to explore the effects of grid size on warm-up time, the spread values over time for the Loam soil with different grids are plotted in Figure 3. The relative difference between the two $t_{wu}$ values is around 4%. Hence, the grid size has insignificant effects on the conclusions of our study.

**[Changes in the manuscript]**

(*previous manuscript*) The flow domain is discretized into 60 grids with a grid size of 5 cm.

(*revised manuscript*) The flow domain is discretized into 60 grids with a grid size of 5 cm, which has been proved to be sufficient for evaluating UIC in our study (results not shown).

*4. Lines 379-383 and Figure 7: The biases of Ks, α, and n as well as their uncertainties differ. Therefore, their RMSEs should not be compared directly. I think it is not a meaningful result that α, which has the largest initial bias and uncertainty, also has the larger RMSE and that n, which has the smallest initial bias and uncertainty, has the smaller RMSE. Their relative improvement might be a better measure.*

**[Response]**

Thank you for your excellent suggestion. To give a fair assessment on the improvement of data assimilation results, we have added the relative error index (*RE*) into the manuscript, which is calculated as,

$$RMSE_m = \sqrt{\frac{1}{N_e} \sum_{j=1}^{N_e} (m_j^E - m^T)^2} \tag{1}$$

$$RE = \frac{RMSE_m^e}{RMSE_m^p} \tag{2}$$

where $m_j^E$ represents the estimated parameter of realization *j* at the last simulation day (EnKF) or the last iteration (IES); $m^T$ represents the true parameter listed in Table 1. $N_e$ is the total number of the realizations. $RMSE_m^e$ and $RMSE_m^p$ represent the $RMSE_m$ of the estimated and prior parameters. Compared with the $RMSE_m$, this index can better present the improvement of parameter estimation during data assimilation. We have modified the figures.

**[Changes in the manuscript]**

[Figure]

**Fig. 8. (revised manuscript)** The *RE* results of parameter estimations (*α*, *n* and $K_s$) under five initialization methods with (a) EnKF and (b) IES.

[Figure]

**Fig. 9. (revised manuscript)** The *RE* results of parameter estimations under five initialization methods with IES when the simulation time is 60 days.

*5. Line 286: You find that $t_{wu}$ is less than one day for sand. I think that this result might be due to the chosen initial condition for the warm-up. It is true, that for the chosen high water contents sand will drain very fast and rapidly approach a similar water content state. However, in case of an initial condition in a very dry state (which should be relevant for the arid climate), the hydraulic conductivity of sand drops to very low values and the initial spread can extend for a very long time, or until a sufficiently large rain event increases the water content and then leads again to the rapid approach of the similar water content. I think it would be interesting to investigate this by choosing a different (dry) initial condition. At least this should be discussed in the manuscript.*

**[Response]**

Thank you for your valuable comment. To further investigate this problem, we conducted another two Monte-Carlo simulations for sandy soil with wet and dry initial conditions (i.e., the mean values of soil moisture ensemble are 0.2375 and 0.15 respectively with the same standard deviation of 3%). The temporal change of the spread $S_p$ and the corresponding $t_{wu}$ are presented in Figure 4.

[Figure]

**Figure 4.** The temporal change of spread for sandy soil with a wet and dry initial condition separately (i.e., the mean of soil moisture ensemble is 0.235 and 0.15 respectively with a standard deviation of 3%).

The results confirm the reviewer's comment: a drier initial condition leads to the increase of warm-up time. Starting with a high soil moisture, the sand drains rapidly; when the soil is very dry, the hydraulic conductivity is extremely small and the initial spread survive for a long time.

**[Changes in the manuscript]**

We have added the discussion about the effects of the mean value of the initial soil moisture ensemble on the warm-up time in Section 3.2.2: "It should be noted that the $t_{wu}$ is also relevant to the initial state of soil. Regarding the initial condition in an extremely dry state under the arid climate, the hydraulic conductivity is very small, and the initial spread extends for a long time. For instance, $t_{wu}$ of sand increases from 1 day to 8 days when the ensemble mean value of initial soil moisture decreases from 0.2375 to 0.15 (results not shown). Yet, if a sufficiently large rain event takes place, the soil moisture increases and then converges to a similar state rapidly." Please see lines 312-316.

*6. Line 273-275 and 477-478: Since you recommend the choice 0.5% as a threshold: Please explain why. What is the advantage? Why should I not choose the other mentioned thresholds (e.g. 1% or 0.1%)?*

**[Response]**

The threshold of 0.5% is recommended due to a reasonable trade-off between the model accuracy and computational cost.

**[Changes in the manuscript]**

An explanation is added in Section 3.2.2: "As shown in Fig. 3(c), there is a significant diversity of the results between Spin-up and Monte-Carlo methods at index value of 1%, indicating that UIC still plays a significant role. In contrast, the requested $t_{wu}$ is more than 15 months for a value of 0.1%. To balance the estimation accuracy and computational cost, we recommend a threshold of 0.5% for both Spin-up and Monte-Carlo methods, and the corresponding warm-up time $t_{wu}$ is 8 months, which is sufficiently long for UIC to diminish and the difference between $PC$ and $S_p$ is insignificant." Please see lines 275-280.

*7. Lines 306-313 and Figure 5: If I understand correctly, you investigate when the uncertainty for the full profile drops below the 0.5% threshold. In addition (or possibly as replacement) I would find it interesting to see the spatially resolved times for each depth for the deepest profile (20 m).*

**[Response]**

Thank you for your valuable comment. We have added a new subfigure (Fig. 5(b)) in the manuscript, which presents the $t_{wu}$ value for each depth with a 20-m soil profile, as presented below. The result still supports our previous conclusion, and more details about UIC along the soil profile have been displayed and analyzed in the revised manuscript.

**[Changes in the manuscript]**

[Figure]

**Fig. 5.** The value of the warm-up time $t_{wu}$. (a) The overall profile $t_{wu}$ values versus different soil profile lengths and (b) $t_{wu}$ value as a function of depth $z$ with a 20-m soil profile.

*8. Line 152 (Equation 8): If the monthly average of the previous year is required, wouldn't that imply that PC is not defined for the first year? However Fig. 3 shows PC starting from time 0. Please clarify.*

**[Response]**

We are sorry that we did not explain it clearly. *PC* is an index that reflects the deviation of soil moisture between two adjacent years in a recursive run after a period of warm-up time $t_{\mathrm{wu}}$. Following de Goncalves et al. (2006) and Ajami et al. (2014), *PC* at month $t=t_{\mathrm{wu}}$ is calculated by comparing the relative difference of soil moistures at month $t=t_{\mathrm{wu}}$ and month $t =t_{\mathrm{wu}} + 12$. As presented in Fig. 3(a) in the manuscript, *PC* at month $t=12$ is close to 0.

**[Changes in the manuscript]**

We have updated Equation 8 as

$$PC(t) = 100 \left| \frac{M(t) - M(t+12)}{M(t+12)} \right| \tag{8}$$

where $M(t)$ and $M(t+12)$ are the monthly averaged soil moistures after model spin-up for $t$ months and $t+12$ months. Please see lines 160-162.

*9. Figure 3: Why does the water content state after 24 months differ between panel (a) and (b)? Since both are initialized with the same parameter values and the UIC has*

*already decayed, they should show the same soil moisture. Please clarify.*

**[Response]**

Good eyes! Thank you for pointing out this problem. We made a mistake when calculating the monthly-average soil moisture at $t = 24$ month in Fig. 3(b). This error is amended in the updated figure.

**[Changes in the manuscript]**

[Figure]

**Fig. 3.** Comparison of Spin-up and Monte-Carlo methods in determining warm-up time. (a) Spin-up method with monthly-averaged soil moisture versus time by running a simulation recursively for 10 years, (b) Monte-Carlo method with monthly-averaged soil moisture of different realizations versus time based on various initial conditions, and (c) Comparison of *PC* and $S_p$ versus time. For the purpose of demonstration, the parameter uncertainty is not considered and we only show the results of the first two years in the figure.

*10. Lines 370-377: Based on Figure 6, I disagree with the statement that filter inbreeding is not a significant issue for the EnKF case. In Figure 6, it seems that the final parameter value does not change any more over time and is over 5 standard deviations away from the truth. This means that the uncertainty is too small. Part of the reason could be that the initial uncertainty is chosen way too small. It is over 9 standard deviations away from the true value. This makes it very difficult for the EnKF to find the true value. I would suggest to repeat the simulations with a larger parameter uncertainty.*

**[Response]**

Thank you for your suggestion. In order to explore the effects of parameter uncertainty on the data assimilation results, we compared the parameter estimations of $\ln K_s$ with

various initial standard deviations of initial parameters (0.1 and 1 respectively), as presented in Figure 5 below. The results agree with our previous conclusion.

(1) We admit that the data assimilation results could be enhanced with a larger parameter uncertainty for the EnKF case, since the parameter updates more rapidly than the small-variance case (Figure below), and could prevent possible inbreeding problems. As shown in the figure, the data assimilation results with a larger parameter uncertainty ($\ln K_s$-1 (HfSatu) and $\ln K_s$-1 (WUP)) are better than those with a smaller one ($\ln K_s$-0.1 (HfSatu) and $\ln K_s$-0.1 (WUP)).

(2) WUP is still the best initialization method among the five approaches, regardless of $\ln K_s$ variance of 0.1 or 1.0. As shown in the figure, the *RMSEs* of $\ln K_s$-1 (WUP) and $\ln K_s$-1 (HfSatu) are 0.13 and 0.36 respectively. Since our topic of this study is to demonstrate the effects of uncertainty of initial condition and initialization methods, we have not revised the model inputs of prior parameters but added the discussions about enlarging the initial parameter uncertainties and filter inbreeding issue in the revised manuscript.

[Figure]

**Figure 5.** The temporal change of estimated mean and standard deviation of $\ln K_s$ with various parameter uncertainties.

**[Changes in the manuscript]**

We have added our discussion about filter inbreeding and initial parameter uncertainties in Section 3.3.2.

*11. Line 177-178: ": : :, uk are state variables (i.e., pressure head and soil*

*moisture*) : : :". *Do you update water content and matric potential of the same node simultaneously in the augmented state? Due to their nonlinear relation, the analysis would lead to inconsistencies between water content and matric potential for the analysis. How is this handled in the forecast? Please clarify.*

**[Response]**

Thank you for your careful reading. In this study, we only update the soil moisture in the simulation and convert the updated soil moisture ensemble into the pressure head to drive the model. The above statement is confusing and we have revised it.

**[Changes in the manuscript]**

(*Previous manuscript*) $\mathbf{u}_k$ are state variables (i.e., pressure head and soil moisture) at time $t_k$.

(*Revised manuscript*) $\mathbf{u}_k$ are the state variables (i.e., soil moisture) at time $t_k$. The updated soil moisture ensemble can be converted to pressure head to drive the model.

**Technical comments:**

*Lines 228-238: This part describes IC-WUP and IC-WUE. However, this is not a general description. In Section 3.2, when the spin up periods are investigated, a different procedure is used. This confused me when reading the paper the first time. Please, only mention the general settings in 3.1 (i.e. climates, soils and model representation), and not specifics that only apply to 3.2 or 3.3. Therefore I would suggest to move this part to Section 3.3. Additionally, here it is not clear how the parameter and initial condition ensembles are exactly generated. Please clarify.*

**[Response]**

Thanks for your valuable suggestions and comments. We are sorry that we did not make the description clear.

IC-HfSatu is a uniform soil moisture profile with the 50% relative saturation (e.g., $\theta=0.254$ loam) of soil; IC-ObsInt is a linear interpolation between observations at the beginning of simulation. The depths of the initial observations are 10 cm, 80 cm, 150 cm, 220 cm and 290 cm; IC-Flux is a steady-state soil moisture profile by warming up the model with a constant infiltration flux (1 mm/d). Besides, the initial conditions of

two warm-up methods are given by running the model prior to the beginning of simulation period with available meteorological data (as shown in Fig. 2). If the meteorological data before the simulation period is available, it is used in the warm-up method to obtain the initial condition (IC-WUP); otherwise, we use the meteorological data at the simulation period (IC-WUE) as a surrogate. The length of warm-up time for IC-Flux, IC-WUP and IC-WUE is equal to $t_{wu}$ (242 days) according to the results in Section 3.2.2(a) , so the warming-up period of WUP for these three methods is from day 124 to day 365. In addition, IC-HfSatu and IC-ObsInt are assumed to be deterministic without uncertainty, while for the IC-Flux, IC-WUP and IC-WUE, the uncertainty of states are introduced by warming up the model with uncertain parameters. The initial realizations of soil hydraulic parameters $K_s$, $\alpha$ and $n$ are generated following logarithm normal distributions, with mean values of 4.7 m d$^{-1}$, 8.6 m$^{-1}$ and 1.8, and variances (log-transformed) of 0.1, 0.3 and 0.006. The saturated soil moisture $\theta_s$ and residual soil moisture $\theta_r$ are assumed to be deterministic with the value of 0.43 and 0.078.

**[Changes in the manuscript]**

We have added the explanation in the manuscript to Section 3.3 as suggested.

*Line 222-226: I think this part should be moved to Section 3.3 as well.*

Thanks. This part has been moved.

*Line 243: "Fig. 1" should be "Fig. 2".*

Thank you. The error has been corrected.

*Line 254 and Fig. 3: The text mentions a simulation length of 10 years, the figure shows only 2 years. I would suggest to mention that you only show the first 2 years.*

Thanks. This has been revised.

*Lines 338-342: How many observations are there? In what depths are the observations? What is the assimilation frequency? Or is only a single observation in the depth of 10*

*cm assimilated every 10 days? If that is the case this has to be clarified.*

**[Response]**

Thank you for your suggestion. The observations are only collected at the depth of 10 cm and assimilated every 10 days, starting from day 3. Unless otherwise specified, the total numbers of the observations are 37 (3rd, 13th, 23th,…, 363th days).

**[Changes in the manuscript]**

(*Previous manuscript*) In addition, the observation at 10 cm is assimilated into model every 10 days.

(*Revised manuscript*) A total number of 37 observations are assimilated into the model. The observation depth is at $z = 10$ cm and the observed soil moisture is assimilated every 10 days, starting from day 3.

*Lines 343-350. I think this part should be moved to methods in Section 2.*

Thanks. Revised.

*Line 352-353 and Figure 6: I would mention that this is case 1 and case 2.*

Thanks. This has been modified.

*Line 399: "Field" instead of "Filed".*

Thank you. Revised.

*Figure 4: Since essentially the times for sand for all climates as well as silt and clay loam for the M-AC and the M-SC climate can not be properly displayed: Maybe the logarithm of the time could be more meaningful (like in Fig. 5).*

**[Response]**

Thank you. Fig. 4 has been revised in the manuscript according to the suggestions from you and reviewer 2. We added a case to investigate the $t_{wu}$ value in a multiple-layers soil.

**[Changes in the manuscript]**

[Figure]

**Fig. 4.** The length of warm-up time $t_{wu}$ with various soils and meteorological conditions. Note that some of the $t_{wu}$ values are larger than 10 years and are not able to be obtained due to the 10-year simulation time. The heterogeneous soil profile consists of Loam (0-75 cm), Clay loam (75-150 cm), Silt (150-225 cm), and Sand (225-300 cm).

*Figure 6: The line for IC-WUE is essentially not visible. Is it below IC-WUP? At least mention this in the caption.*

**[Response]**

Thank you for pointing out this problem. We are sorry that we have used mistaken model input folder when analyzing the data from WUE and WUP, and two identical curves for WUE and WUP were generated. The error has been fixed, and the figure has been updated. Although there is a little difference between Figs. 6 (a) and (b), the general conclusions are consistent with the previous ones.

**[Changes in the manuscript]**

[Figure]

**Fig. 6.** The results of ln$K_s$ estimations (first row) and their associated standard deviations (second row) within two data assimilation frameworks (left: EnKF; right: IES) under five initialization methods.

**[Response]**

Thank you for your comment, according to the suggestions from you and reviewer 1,

we have added a new figure to explore the effects of ensemble size on the parameter estimations within EnKF and IES.

**[Changes in the manuscript]**

[Figure]

**Fig. 7. (revised manuscript)** The impacts of increased ensemble size and the uncertainty of initial state on the results of $\ln K_s$ estimations within EnKF and IES.

The results of IC-HfSatu-500 and IC-ObsInt-500 with the ensemble size of 500 in Fig. 7 are similar with those of IC-HfSatu and IC-ObsInt (Fig. 6), indicating that the improvement of the parameter estimation result is slight when the ensemble size increases from 300 to 500. Hence, the ensemble size of 300 is sufficient for data assimilation problem in this study.

*2) The synthetic case study present the proper warm-up time $t_{wu}$ versus different soil texture, soil depth, and meteorological conditions. While the relationship between $t_{wu}$ and meteorological conditions may be commonsense, the reveal of quantitative relationship between $t_{wu}$ and soil texture and soil depth is surprising and interesting, due to the fact that $t_{wu}$ changes abruptly from sand to finer texture, and it increases nonlinearly with the increase of soil depths. However, the soil is seldom homogeneous in natural conditions, especially for very long soil profile. The authors should at least present one simulating result of $t_{wu}$ for layered soil, which is more applicable for real-world case. I believe this should take too much work since it is one-dimensional model.*

**[Response]**

Thank you for your valuable comment. We have added a case to obtain the $t_{wu}$ for a layered soil profile, which consists of Loam (0 to 75 cm), Clay loam (75 to 150 cm), Silt (150 to 225 cm) and Sand (225 to 300 cm).

**[Changes in the manuscript]**

[Figure]

**Fig. 4.** The length of warm-up time $t_{wu}$ with various soils and meteorological conditions. Note that some of the $t_{wu}$ values are larger than 10 years and are not able to be obtained due to the 10-year simulation time. The heterogeneous soil profile consists of Loam (0-75 cm), Clay loam (75-150 cm), Silt (150-225 cm), and Sand (225-300 cm).

In the revised manuscript, we have expanded the results and presented the $t_{wu}$ value in layered soil: "Moreover, regarding the heterogeneous soil with multiple layers, the $t_{wu}$ under the M-AC is larger than 10 years (similar to Silt and Clay loam), while that under M-SC or M-HC becomes much smaller (higher than that of Loam but they are of the same magnitude). Thus, it is conjectured that $t_{wu}$ is determined by the fine soil texture in the layered profile under dry meteorological condition, but averaged soil hydraulic properties under wet meteorological condition". Please see lines 307-311.

**Minor comments:**

*Line 12: various initial condition »> various initial conditions*

Thank you. Revised

*Line16: model initializing »> model initializing methods*

Thank you. This has been revised.

*Line 28: delete in*

Thank you. Revised.

*Line 48: a space between approaches and comma*

Thanks. The error has been corrected.

*Line 61: hereafter referred »> hereafter referred to*

Thank you. It is revised

*Line 77: delete the last the*

Thank you. It is modified.

*Line 81: initial ensemble are »> initial ensemble is*

Thank you. It is rewritten.

*Line83: Currenlty »> Currently*

Thank you. Revised

*Line 110: Richards's »> Richards'*

Thanks. Revised

*Lines 129-130: as state dependent, atmospheric boundary condition (try to be more concise here and some other statements)*

Thanks. It is rewritten.

*Line 135: detemined »> determined*

Thanks. Revised

*Line 141: use UIC instead Eqs. (9-10): try to use one equation instead and shorten the description of the equation.*

Thank you. It is revised.

*Line 172: assimilation + approach*

Thanks. Revised

*Line 210: which λ values you use in the simulations?*

Thanks. $\lambda$ is a dynamic stability multiplier during the iterations. The prior value of $\lambda$ is 10, but the value can be adjusted adaptably according to the data assimilation results at every iteration.

*Line 222: perscribe »> prescribe*

Thank you. It is revised.

*Line 223: availablitity »> availability*

Thanks. Revised.

*Line 256: be consistent using italic or not for PC.*

Thank you. This has been revised.

*Line 256: why 3%?*

Thank you. The Gaussian noise is determined as 3% according to the observation error of soil moisture since the uncertainty of parameter is not taken into consideration in this part. We have added a sentence to explain it.

*Line 335: warms »>warm*

Thanks. Revised.

*Line 356: delete both*

Thanks. This has been revised.

*Line 358-359: thus »> and thus*

Thanks. Revised.

*Line 372: multiple spaces between runs and are.*

Thanks. Revised.

*Change "than" to that*

Thanks. This has been revised.

*Line 405: Which evapotranspiration model are you using?*

Thank you for your comment. The potential evaporation is calculated by Penman-Monteith's equation. We will add an explanation in the manuscript.

*Line 427: needs a space after "part."*

Thanks. Revised.

*Lines 443-444: "soil moisture profile has large variation, e.g., discontinuous soil moisture in layered soils." — it would be interesting to see an additional case for heterogeneous soils, and this also leads to another interesting question — what will happen if pressure head profile, which is continuous in heterogeneous soil, is used as initial condition. Please add some discussion on this topic.*

**[Response]**

Thank you for your valuable comments. We have added the case about $t_{wu}$ of layered soil in the manuscript, please see Fig. 4 above. Regarding the topic about using initial pressure head as initial condition, we are going to discuss it from three aspects:

(1) It is easier to collect the soil moisture data than soil pressure head, so that we only

use soil moisture as observation in this study. In heterogeneous soil, the pressure head profile is continuous compared to soil moisture profile, which is the reason why head is preferred as the state variable in numerical models.

(2) With respect to $t_{wu}$, the conversion relationship between the spread of soil moisture and pressure head is deterministic (i.e., the spread as well as the $t_{wu}$ value with pressure head profile can be derived from those with soil moisture profile). Thus, in this study we choose soil moisture in the study of UIC and $t_{wu}$.

(3) The impact of observation type (i.e., pressure head and soil moisture) on data assimilation results has been widely explored in previous studies (Shi et al. 2015), since these two state variables have different probability distributions, nonlinearity, accessibility, and observation errors. Although choice of head or moisture can affect the data assimilation results, it is not primarily induced by the difference in UIC, which is focus of current study.

**[Changes in the manuscript]**

We have added some discussions about the effects of initial pressure head profile in heterogeneous soil in Section 5. Please see lines 545-552.

*Line 452: atmospheric condition »> atmospheric boundary condition*

Thanks. Revised.

*Conclusion 2: Please include more details and add quantitative conclusions for this.*

**[Response]**

Thank you. We have modified the conclusion 2 according to your suggestion.

**[Changes in the manuscript]**

Warm-up time varies nonlinearly with soil textures, meteorological conditions, and soil profile. Under most situations (e.g., Loam with the soil profile length less than 5 m under non-arid climate), one-year warm-up time is sufficient for soil water movement modeling, but an extremely long time (exceeds 10 year) is needed to warm up the model for a long, fine-textured soil profile under an arid meteorological condition.

*Errors in references: Line 566, Line 673, Line 610, Line 639.*

Thanks. Revised.

**[Response]**

In this study, we conducted both synthetic and field experiments. In synthetic experiments the "observed" data are generated by running the forward model with the exactly known parameters, while the field data are collected in the experimental station. The field observations may contain a lot of uncertainties, such as unknown/inaccurate upper and lower boundary conditions, unknown observation error/bias, and unknown parameters. All these unknown uncertainties have impact on the modeling, while the effect from UIC could be overshadowed by those from other unknown uncertainties, and the direct results from field experiments could by inconclusive. By utilizing the synthetic observations, we can separate the effects on modeling from all these uncertainties, since they are all perfectly known. Based on the synthetic case, we can elaborate our conclusions on the temporal evolution of UIC, as well as its effect on data assimilation, which is assessed by the estimated parameter and perfectly-known true parameter. We think this is the logic way: first, a comprehensive investigation on UIC is conducted using synthetic case, then, the field data can be used to validate the applicability of our approaches/results.

The conclusions by using field and synthetic data are similar (the difference of results between various initialization methods are less significant in field case, due to contamination of other unknown uncertainties), indicating a good applicability of our approaches/results.

**[Changes in the manuscript]**

We have added an explanation "Synthetic observation in previous section is generated by running the model with exactly known uncertainty sources. By conducting synthetic experiments, we can thoroughly analyze the impact of UIC during data assimilation, with scenarios having different numbers of observations/unknown parameters, and more decisive conclusions can be drawn. In contrast, the field observations contain

additional uncertainties which are largely unknown (e.g., the calculated evapotranspiration is inaccurate for real-world case). In order to examine the real-world applicability of the conclusions drawn from synthetic case, Field data are necessary to validate our results." in the manuscript. Please see lines 462-468.

*5. Please describe more details about the novelty of this paper, it seems there is no new method involved, and I am not sure how useful and novel the conclusions are.*

**[Response]**

Although the initialization and data assimilation methods used in this study are not new, we claim the innovation of our studies mainly based on that, to the best of our knowledge, this is the first study analyzing the effects of initial conditions and initialization methods within various data assimilation frameworks to date. The novelties include three aspects.

(1) Two common approaches (Spin-up method and Monte Carlo method) for quantifying the temporal evolution of initial condition uncertainty are compared. Spin-up methods are widely used in large-scale hydrological model due to their smaller computational cost. However, Monte Carlo methods have the merit that they can explicitly quantify UIC, which is suitable for data assimilation. Finding an equivalency between these two methods can fill the gap between widely-used initialization methods and data assimilation, both of which are important tools increasing the accuracy of hydrological modeling. Also, new algorithm is not necessary for the combination of UIC/initialization and data assimilation, in both of which we use Monte Carlo method for expressing state uncertainty. This (i.e., no new algorithm) should be regarded as an advantage according to principle of parsimony, since our approach can be easily applied and validated by the readers.

(2) The influences of soil texture, meteorological condition and soil profile length on initial condition uncertainty evolution are exploited. Especially, we propose a warming-up time $t_{wu}$, which is defined as the time when percentage cutoff $PC$ or ensemble spread $S_p$ is lower that 0.5%, can guide us to select the warming up period and pick up observation at different time in data assimilation.

(3) Different approaches to initialize unsaturated-saturated flow models within two data assimilation frameworks (IES and EnKF) are assessed. Our studies focus on the case when both parameters and initial state are uncertain, and the combination of selected initialization method and data assimilation can be a standard approach for future variably saturated flow modeling.

**[Changes in the manuscript]**

We have modified our manuscript according to the discussion above to make the novelties of the paper more apparent.

[revised manuscript text omitted]

$$\frac{\partial h}{\partial z}\bigg|_{z=z_N}=0 \quad \frac{\partial h}{\partial z}\bigg|_{z=z_N}=0 \tag{7}$$

where $z_N$ is the depth of bottom boundary.

**2.2 UIC quantification**

155     The investigation of uncertainty in this study includes model states (e.g., soil moisture) and model

parameters, where UIC is a special case of state uncertainty

 at $t=0$. The analysis is twofold. First, we consider a particular situation when

UIC is the only uncertain source and all the model parameters are known. Thus, the choice of initial conditions is solely responsible for the accuracy of the model outputs. In this case, the temporal decay of UIC (either with or without observation) can be clearly demonstrated. by utilizing Spin-up or Monte-Carlo methods. Second, a more complex and realistic situation, including both uncertain initial condition and model parameters, is considered during the data assimilation of soil moisture observation. Data assimilation methods, i.e., EnKF and IES, are used to update the model parameters and state simultaneously when observation data are availableUIC and data assimilation are smoothly combined in our approach since we choose Monte-Carlo-based methods (EnKF and IES). At $t$=0, we generate an ensemble of soil moisture profiles based on one initialization method (which introduces UIC), and use this ensemble to initiate the data assimilation (assimilate observations and estimate parameter). Finally, we can evaluate our data assimilation performance based on different initializing methods.

2.2.1 The indexes of Spin-up and Monte-Carlo methods

The uncertainty of initial condition can be measured by the percent change $PC$ for Spin-up method (Ajami et al., 2014; Seck et al., 2014) or the ensemble spread $S_p$ for Monte-Carlo method (Reichle and Koster, 2003). Percent change isPercent change is an index that reflects the deviation of soil moisture between two adjacent years in a recursive run after a period of warm-up time $t_{wu}$, which could be calculated as:

$$PC = 100\left|\frac{M_1 - M_2}{M_2}\right| \quad PC(t) = 100\left|\frac{M(t) - M(t+12)}{M(t+12)}\right| \tag{8}$$

where $M_1$ is$M(t)$ and $M(t+12)$ are the monthly mean ofaveraged soil moisture from the previous year and $M_2$ is the monthly averagemoistures after model spin-up for the current year$t$ months and $t$+12 months (de Goncalves et al., 2006).

The ensemble spread ($S_p$)), calculated as a square root of averaged variance over all interested nodes, is an useful index to quantify the difference ofamong various realizations in Monte Carlo simulation, and it is given as:

$$S_p(k) = \sqrt{\frac{1}{N}\sum_{i=1}^{N} Var(y_{i,k})} \quad S_p(k) = \sqrt{\frac{1}{N(N_e-1)}\sum_{i=1}^{N}\sum_{j=1}^{N_e}(y_{i,j\,k}^{a} - \langle y_{i\,k}^{a}\rangle)^2}$$

$$\tag{9}$$

$$\tilde{Var(y_{i,k})} = \frac{1}{N_e - 1}\sum_{j=1}^{N_e}(\mathbf{y}_{i,j,k}^a \quad \langle \mathbf{y}_{i,k}^a\rangle)^2 \tag{10}$$

where   $y_{i,j,k}^a$ is nodal soil moisture value;  $\langle y_{i,k}^a\rangle$ is the ensemble mean of  $y_{i,j,k}^a$; $i = 1, 2, \ldots, N$ are the nodes of interest (can be part of the profile); $j=1, 2, \ldots, N_e$ is the ensemble number index; $N_e$ is the ensemble size, which is taken as 300 in this study based on sensitivity analysis of the ensemble size on the calculated results.  When $N = 1$, the concept of $S_p(k)$ is equivalent to the standard deviation of  $y_k^a$ at one location and time $t_k$.

2.2.2 Data assimilation approaches

We employ EnKF and IES for data assimilation  in this study. Fig. 1 illustrates the basic ideas and differences of the two methods.

EnKF approach was first proposed by Evensen (1994) and has been widely used in variably saturated flow problems (Huang et al., 2008; De Lannoy et al., 2007). This approach is a sequential data assimilation method (as shown in Fig. 1(a)) which incorporates observations into the model in order.

In this part, we assume that hydraulic parameters $K_s$, $\alpha$, and $n$ are unknown, while the other parameters $\theta_r$ and $\theta_s$ are  deterministic. The vector of parameter and state is described as,

$$\tilde{\mathbf{y}_k} = [\mathbf{m}_k, \mathbf{u}_k]^T \tag{11}$$

$$\mathbf{y}_k = [\mathbf{m}_k, \mathbf{u}_k]^T \tag{10}$$

where  $\mathbf{m}_k$ is the parameter vector (i.e., $K_s$, $\alpha$, and $n$),  $\mathbf{u}_k$ are state variables (i.e.,  soil moisture) at time $t_k$, the dimension of $y_k$ is $N_y$: $N_y = N_m +  N_d$, where $N_m$ indicates the amount of the parameters to be estimated; $N_d$ are the number of nodes of the numerical model. The updated soil moisture ensemble can be converted to pressure head to drive the model. The observation vector can be defined as,

$$\cancel{\mathbf{d}_{j,k} = \mathbf{d}_k + \mathbf{\varepsilon}_{jk}} \qquad (12)$$

$$\underline{\mathbf{d}_{j,k} = \mathbf{d}_k + \mathbf{\varepsilon}_{j,k}} \qquad (11)$$

where $\cancel{\mathbf{d}_k}\,\underline{\mathbf{d}_k}$ denotes the observation at time $t_k$; $\cancel{\mathbf{\varepsilon}_{jk}}\,\underline{\mathbf{\varepsilon}_{j,k}}$ ($j$=1, 2, …, $N_e$) are independent Gaussian noises added to the observations; $\cancel{\mathbf{d}_{j,k}}\,\underline{\mathbf{d}_{j,k}}$ is the observation vector for ensemble index $j$ at time $t_k$. Based on the differences of model forecast and observations, the state-parameter vector can be updated as:

$$\cancel{\mathbf{y}_{j,k}^a = \mathbf{y}_{j,k}^f + \mathbf{K}_k(\mathbf{d}_{j,k} - \mathbf{H}\mathbf{y}_{j,k}^f)} \qquad (13)$$

$$\underline{\mathbf{y}_{j,k}^a = \mathbf{y}_{j,k}^f + \mathbf{K}_k(\mathbf{d}_{j,k} - \mathbf{H}\mathbf{y}_{j,k}^f)} \qquad (12)$$

where $\cancel{\mathbf{y}_{j,k}^f}\,\underline{\mathbf{y}_{j,k}^f}$ denotes the estimated or initially guessed values of parameter and state, while $\cancel{\mathbf{y}_{j,k}^a}\,\underline{\mathbf{y}_{j,k}^a}$ is the updated estimates; $\mathbf{H}$ is an observation operator, linking the relationship between the state-parameter vector and the observation vector. $\cancel{\mathbf{K}}\,\underline{\mathbf{K}}$ represents the Kalman gain matrix, which can be calculated as,

$$\mathbf{K}_K = \mathbf{C}_k^f \mathbf{H}^T [\mathbf{H}\mathbf{C}_k^f \mathbf{H}^T + \mathbf{C}_{D_K}]^{-1} \qquad (\cancel{14}13)$$

where $\cancel{\mathbf{C}_{D_k}}\,\underline{\mathbf{C}_{D_k}}$ indicates the covariance matrix of observed data errors, while $\cancel{\mathbf{C}_k^f}\,\underline{\mathbf{C}_k^f}$ is the error covariance matrix of forecast ensemble, given by

$$\cancel{\mathbf{C}_k^f \approx \frac{1}{N_e-1}\sum_{j=1}^{N_e}\left\{\left[\mathbf{y}_{j,k}^f \langle\mathbf{y}_k^f\rangle\right]\left[\mathbf{y}_{j,k}^f \langle\mathbf{y}_k^f\rangle\right]^T\right\}} \qquad (15)$$

$$\underline{\mathbf{C}_k^f \approx \frac{1}{N_e-1}\sum_{j=1}^{N_e}\left\{\left[\mathbf{y}_{j,k}^f - \langle\mathbf{y}_k^f\rangle\right]\left[\mathbf{y}_{j,k}^f - \langle\mathbf{y}_k^f\rangle\right]^T\right\}} \qquad (14)$$

where $\cancel{\langle\mathbf{y}_k^f\rangle}\,\underline{\langle\mathbf{y}_k^f\rangle}$ is the ensemble mean of $\cancel{\mathbf{y}_k^f}\,\underline{\
[revised manuscript text omitted]

940

---

## Author Response (AR2)

Comments by the Associate Editor:

**Editor Decision**

The Editor has decided that some corrections are necessary before the manuscript can be published. We kindly ask you to upload the files required for the production process in your File Manager no later than 21 Jun 2019. Please note that all Referee and Editor reports, the author's response, as well as the different manuscript versions of the peer-review completion (post-discussion review of revised submission) will be published along with your paper in HESS.

**[Response]**

Thanks for handling our manuscript.

Reply to comments from Anonymous Referee #1

**General Comments**

*The manuscript has improved significantly. I highly appreciate the effort of the authors. All my comments have been answered sufficiently enough. I recommend publication.*

**[Response]**

Thanks for your positive comments! We have improved the manuscript according to your suggestions.

**Minor comment:**

*The new case 3 investigates, to which extend adding uncertainty to IC-HfSatu and IC-ObsInt improves the subsequent data assimilation. To do this Gaussian noise is added to the initial conditions.*

*Please clarify, how this is done. Is the noise added to each individual node independently? If this is the case, the added uncertainty will diminish very rapidly due to the dissipative component of the Richards equation. This reduces the added uncertainty strongly and consequently unnecessarily limits the findings.*

*I then recommend to either add a spatially correlated noise (with correlation length of e.g. 20 cm or larger), or to create spatially correlated states by adding the noise to the entire profile (for HfSatu) or to the measurements before interpolating them (in the case of ObsInt). I think, this would strengthen the found results.*

**[Response]**

Thanks for your valuable suggestion. For the initial condition IC-ObsInt-500-Un, we added the noise to individual node independently, as the reviewer pointed out. For initial condition IC-HfSatu-500-Un, we added a Gaussian noise to the 50% relative saturation value (e.g., a soil moisture value of 0.254 for Loam) to create 500 uniform soil profiles (i.e., the soil moisture profile is always uniform for all realizations, but the values differ, which follow the Gaussian distribution with mean 0.254), which implicitly assume an infinite spatial correlation scale of the added noise.

In order to investigate the influences of spatial correlation of the added noise, we add an additional scenario for IC-HfSatu by adding a noise to the entire profile with a standard deviation of 0.017 and correlation length of 50 cm (hereafter referred to as IC-HfSatu-500-Un-SC). In contrast, the correlation scale of the noise in the original scenario IC-HfSatu-500-Un is much larger than 300 cm. The results of IC-HfSatu-500-Un and IC-HfSatu-500-Un-SC are compared in the figure below.

[Figure]

**Figure 1.** The results of $\ln K_s$ estimation and their associated standard deviations with IC-HfSatu-500-Un and IC-HfSatu-500-Un-SC.

As presented in the figure 1, the estimated results of $\ln K_s$ value and standard deviations with IC-HfSatu-500-Un-SC are similar to those with IC-HfSatu-500-Un, indicating that the effects of spatial correlation of noise during initial condition construction (HfSatu and ObsInt) on parameter estimation is not significant in this study, although IC-HfSatu-500-Un-SC has slightly faster dissipation of UIC due to a less spatially-correlated noise during UIC construction.

We present a brief discussion on the influence of spatial correlation of the added noise for (HfSatu and ObsInt) in the revised manuscript.

**[Changes in the manuscript]**

We have added the content "Besides, we also investigate the influences of spatial correlation of the added noise (e.g., with correlation length of 50 cm or infinity) for constructing IC-HfSatu and IC-ObsInt, but their parameter estimation results are similar (not shown), indicating that the effects of spatial correlation of noise during the

construction of IC-HfSatu and IC-ObsInt are not significant on parameter estimation."

Please see lines 422-425.

**Technical comment:**

*Fig. 5: The y axis of the panel (b) is probably wrong (up to 800 years). Maybe the same axis as for panel (a) can be used to be able to compare.*

**[Response]**

Good eyes! The unit of y axis of panel (b) should be "day" rather than "year" in the previous manuscript. We have corrected it in Fig. 5(b) in the revised manuscript.

**[Changes in the manuscript]**

[revised manuscript text omitted]